# Bayesian Meta-Learning for Few-Shot 3D Shape Completion

## Abstract

Estimating the 3D shape of real-world objects is a key perceptual challenge. It requires going from partial observations, which are often too sparse and incomprehensible for the human eye, to detailed shape representations that vary significantly across categories and instances. We propose to cast shape completion as a Bayesian meta-learning problem to facilitate the transfer of knowledge learned from observing one object into estimating the shape of another object. To facilitate the learning of object shapes from sparse point clouds, we introduce an encoder that describes the posterior distribution of a latent representation conditioned on the sparse cloud. With its ability to isolate object-specific properties from object-agnostic properties, our meta-learning algorithm enables accurate shape completion of newly-encountered objects from sparse observations. We demonstrate the efficacy of our proposed method with experimental results on the standard ShapeNet and ICL-NUIM benchmarks.

## 1 Introduction

The task of estimating 3D geometry from sparse observations, commonly referred to as *shape completion*, is a key perceptual challenge and an integral part of many mission-critical problems, including robotics (Varley et al., 2017) and autonomous driving (Giancola et al., 2019; Stutz & Geiger, 2018). Recently, a series of methods (Mescheder et al., 2019; Park et al., 2019) have achieved great success by using the observations to infer the parameters of an implicit 3D geometric representation of the targets object. However, with some notable exceptions (Yuan et al., 2018), such methods require relatively dense observations to achieve high accuracy, which is usually impractical in real situations. In this paper we introduce a novel methodology that enables state-of-the-art shape completion of previously unseen objects from highly sparse observations.

Our insight comes from the following simple intuition: "Can we leverage the geometric information available in one object to improve shape completion results on another target object?" Meta-learning is an emerging field of study in machine learning that serves this very purpose. By training a model on multiple inter-related tasks, it *learns how to learn* new tasks efficiently from a small amount of observations. Recently proposed meta-learning methods often achieve this by parameterizing the input-output relationship with *task-specific* latent variables and training a separate, *task-agnostic* model/mechanism that can infer these task-specific variables from sparse observations of the target task (Chang et al., 2015; Finn et al., 2017; Garnelo et al., 2018). We can cast the shape completion problem as a Bayesian meta-learning problem by treating each object as a *task* and its sparse observations as the corresponding *contextual dataset*.

In popular Bayesian variants of meta-learning (Edwards & Storkey, 2017; Eslami et al., 2018; Garnelo et al., 2018), the task-specific latent variables are treated as *random* variables, and the aforementioned task-agnostic model (i.e. the encoder) is represented as a posterior distribution of the latent variables conditioned on sparse observations. In this study, we combine probabilistic meta-learning with recent shape completion methods that represent the geometry of a given object with implicit parameters, such as the parameters of a signed distance function (SDF). By training an *encoder* that computes the posterior distribution of these implicit parameters conditioned against sparse observations, we develop a framework that enables the few-shot learning of implicit geometric functions. Under appropriate regularity conditions, the computation of correct posterior distribution leads to optimal prediction in the sense of Bayes Risk (Maeda et al., 2020). Our proposed approach is a natural extension of many

implicit approaches, in the sense that it introduces an additional encoder function that represents the posterior distribution of the geometry-describing implicit parameters.

More specifically, we build upon the Bayesian approach of (Maeda et al., 2020) whose posterior estimate behaves asymptotically well with respect to the size of contextual dataset, and combine their method with Implicit Geometrical Regularization (IGR) (Gropp et al., 2020). We use IGR as the baseline, and demonstrate its efficacy on two benchmark datasets (ShapeNet and ICL-NUIM), especially when the observations are very sparse.

## 2 RELATED WORK

### 2.1 3D DECODING REPRESENTATIONS

Unlike images, that contain a clear pixel-based structural pattern, there is no unified representation for 3D object reconstruction that is both computationally and memory efficient. In terms of used 3D representations, existing methods can be broadly divided into the following categories:

**Voxel-based** methods are a generalization of 2D pixels into 3D space, and thus constitute a natural extension for classical image-based methods. Early works focused on 3D convolutions operating on dense grids (Choy et al., 2016) to generate an occupancy function that determines whether each cell is inside an object or not, however these were limited to relatively small resolutions. To address the high memory requirements of dense voxel grids, various works have proposed 3D reconstruction in a multi-resolution fashion (Häne et al., 2017), with the added complexity of requiring multiple passes to generate the final output. More recently, OccNet (Mescheder et al., 2019) proposes encoding a 3D description of the output at infinite resolution, and shows that this representation can be learned from different sensor modalities.

**Signed Distance** methods are an alternative to occupancy functions, where instead of the occupancy state we learn a function describing the signed distance to the object surface (Dai et al., 2017; Stutz & Geiger, 2018). This approach builds upon earlier fusion methods that utilize a truncated signed distance function (SDF) introduced in (Curless & Levoy, 1996). DeepSDF (Park et al., 2019) represents 3D space as a continuous volumetric field, and requires at training time the ground-truth SDF calculated from dense input data using numerical methods. Implicit Geometric Reconstruction (IGR) (Gropp et al., 2020) is a SDF variant that uses Eikonal regularization, thus enforcing that the output of the decoder will be the SDF of "some" surface. This is an effective way of mitigating the impact of outliers in the final generated surface, and is used as the starting point for our proposed meta-learning approach to shape completion.

**Point-based** methods directly output points located on the object surface, thus eliminating the need for a dense representation of the 3D space. Earlier works such as PointNet (Charles et al., 2017; Qi et al., 2017) combined fully connected networks with a symmetric aggregation function, thus achieving permutation invariance and robustness to perturbations. (Fan et al., 2017b) introduced point clouds as a viable output representation for 3D reconstruction, and (Yang et al., 2017) proposed a decoder design that approximates a 3D surface as the deformation of a 2D plane. Point Completion Network (PCN) (Yuan et al., 2018) is a recent architecture that enables the generation of coarse-to-fine shapes while maintaining a small number of parameters. However, a common limitation of all these methods is that they do not describe topology, and thus are not suitable for the generation of watertight surfaces. Also, to change the number of output points, methods like PCN have to re-train their networks entirely, while SDF-based methods learn the geometry in an implicit form and thus can generate any amount of points.

**Mesh-based** methods choose to represent classes of similarly shaped objects in terms of a predetermined set of template meshes. First attempts focused on graph convolutions alongside the mesh's vertices and edges (Guo et al., 2015), and more recently as a direct output representation for 3D reconstruction (Kanazawa et al., 2018). These methods, however, are only able to generate meshes with simple topologies (Wang et al., 2018), require a reference template from the same object class (Kanazawa et al., 2018) and cannot guarantee water-tight closed surfaces (Groueix et al., 2018). A learnable extension to the Marching Cubes algorithm (Lorensen & Cline, 1987) has been proposed in (Liao et al., 2018), however this approach is limited by the memory requirements of the underlying 3D voxel grid.

## 2.2 Encoding and Decoding Mechanism

In shape completion methods, one must construct a geometry of object from point clouds of varying size that are scattered over various regions. Thus, all methods use some mechanism to *encode* the sparse point clouds to a tensor of fixed size, and *decode* this tensor to produce the final output. Existing methods also differ by the design of these decoding and encoding mechanisms.

Methods like DeepSDF and IGR train an auto-*decoder*, and do not train separate encoder functions at all. To find the geometry-describing implicit parameters for each object, these methods apply likelihood-based gradient descent on randomly initialized latent variables, using just sparse observations from the target object. Thus, these methods by design do not use observations from multiple objects to train an *object-agnostic* mechanism that can efficiently learn the latent variables. Meanwhile, PCN and OccNet both train a complex encoder function that maps sparse observations to latent variables. Although their decoders differ entirely (PCN directly outputs 3D points, while OccNet outputs binary values), they both use an encoder that aims to capture the hierarchical structure of the object's geometry. More specifically, PCN's encoder is equipped with a mechanism to extract information from sparse observations in two steps, one aimed at extracting global information and another at extracting local information. On the other hand, OccNet's encoder is a version of PointNet (Charles et al., 2017) that uses max pooling with respect to the sparse observation set.

Our method also uses an encoder function, however it differs from those mentioned above in the sense that it represents a posterior distribution, rather than a deterministic function. Under an appropriate set of regularity conditions , inference made from the predictions of posterior distribution is optimal in terms of the Bayes risk (Maeda et al., 2020). Furthermore, because our encoder is probabilistic, it is capable of outputting multiple *candidate* shapes for a given sparse observation.

## 3 Meta-Shape completion

### 3.1 Problem Setting

Let $D_k = \{x_n^{(k)} \in \mathbb{R}^3 | n = 1, \cdots, N_k\}$ be an arbitrary set of points located on the surface of a 3D object $k$. The goal of shape completion for object $k$ is to use $D_k$ to infer its surface. There are various approaches to this problem: if the surface is closed, one can use an occupancy function $F_{occ}(x)$ that tells whether a point $x$ is inside or outside the surface. One may also use a signed distance function $F_{sdf}(x)$ that evaluates how far point $x$ is to the surface, with positive values indicating that it's outside and negative values indicating that it's side. In both approaches, the transcribed goal is to use $D_k$ to find the object-$k$-specific function $F_k$ that best describes its geometry. Our proposed method focuses on the latter approach, and in the following sections we show how Bayesian meta-learning can be used to learn a probabilistic SDF for shape completion.

### 3.2 Meta-Learning for Shape Completion

Meta-learning exploits underlying similarities among tasks to enable the transfer of knowledge between tasks, so that information gained from solving one problem can improve performance on another. Traditional meta-learning methods work by parameterizing their models with two types of parameters: *task-specific* and *task-agnostic*. Following this basic philosophy, we represent the signed distance function (SDF) of all observed objects using *object-specific* parameters $h_k$ and *object-agnostic* parameters $\theta$. More specifically, we write our SDF for object $k$ as $SDF_\theta(x; h_k)$, and determine $h_k$ using the $\theta$-parameterized function that encodes $D_k$. The function that maps $D_k$ to $h_k$ is often referred to as *encoder* in the meta-learning literature. The SDF, on the other hand, plays the role of *decoder*. In a probabilistic setting, we consider a probabilistic SDF of form $\mathcal{N}(s|SDF_\theta(x; h_k), \sigma_\theta^2)$.

Now, if we interpret this estimation problem as a case of minimizing the Bayes risk (Maeda et al., 2020), we can show that the optimal solution is given by the predictive distribution $p_\theta(s|x; D_k)$ computed using the posterior distribution $p(h|D_k)$:

$$p(s|x, D_k) = \int p(s|x, h_k)p(h_k|D_k)dh_k. \tag{1}$$

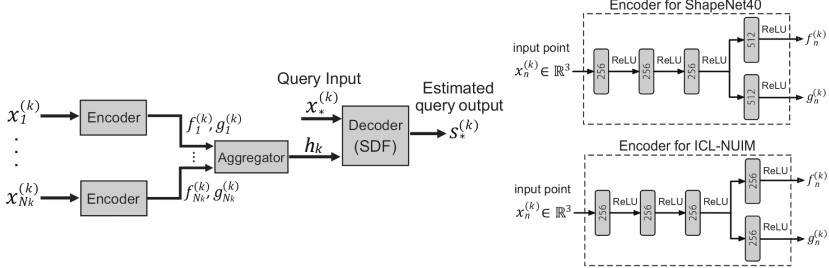

Figure 1: Schematic of our proposed SDF-based meta-shape completion method.

The estimation of the posterior $p(h_k|D_k)$ is a challenging task: for the approximation to be valid, it must be able to accept unordered sets $D_k$ of various lengths, while satisfying all other properties of a posterior distribution. For example, its variance must approach zero as we take the number of observed points to infinity. According to the theory of (Maeda et al., 2020), we can construct an approximation that satisfies all these requirements using a Gaussian distribution $p_\theta(h_k|D_k) = \prod_{i=1}^{d} \mathcal{N}(h_{k,i}|\mu_{i,\theta}(D_k), \sigma_{i,\theta}^2(D_k))$ with mean and variance as follows (we use $h_{k,i}$ to denote the $i$-th element of $h_k$):

$$\mu_{i,\theta}(D_k) = \sigma_{i,\theta}^2(D_k) \sum_{n=1}^{N_k} \frac{f_{i,n}}{g_{i,n}^2}, \quad \sigma_{i,\theta}^2(D_k) = \left( \sum_{n=1}^{N_k} \left( \frac{1}{g_{i,n}^2} - \frac{1}{g_{i,0}^2} \right) + \frac{1}{g_{i,0}^2} \right)^{-1}. \quad (2)$$

In the above, $f_{i,n} = f_{i,\theta}(x_n^{(k)}, y_n^{(k)})$ and $g_{i,n} = g_{i,\theta}(x_n^{(k)}, y_n^{(k)})$ are neural networks parameterized by $\theta$. When making a probabilistic inference, we sample from this Gaussian posterior approximation and feed it to the decoder SDF. A schematic diagram of our proposed meta-learning shape completion method can be found in Figure 1.

The training proceeds by making predictions about target points using contextual information from various datasets. To prepare the *mock*-target points at training time, we decompose $D_k$ into $D_k^{ctxt}$ and $D_k^{targ}$ for each task $k$ and use the following classic ELBO (Ranganath et al., 2014; Kingma & Welling, 2014) to optimize the predictive distribution given $D_k$:

$$\mathcal{L}_k(\theta) := - \int p_\theta(h_k|\mathcal{D}_k) \left( \sum_{n=1}^{N_k} \log p_\theta(s_n^{(k)}|x_n^{(k)}, h_k) + \log p_\theta(h_k|D_k^{ctxt}) \right) dh_k - H(p_\theta(h_k|\mathcal{D}_k)). \quad (3)$$

Here $s_n^{(k)}$ is the true signed distance from point $x_n^{(k)}$ to the surface of object $k$. In our problem setting, $s_n^{(k)}$ is 0 for all $(n, k)$ because all the observed points are on the surface of the objects. Figure 1 contains a diaram of our proposed algorithm. To test the sheer efficacy of our meta-approach, we only use MLP for encoder in this study, and do not explore the specific architecture that is suited for the shape-completion problem.

### 3.3 EIKONAL REGULARIZATION

The datasets $D_k$ considered in shape completion tasks usually only contains points on the object surface. However, it is difficult to enforce the learned function to be a *signed distance function* (SDF) just by enforcing that is value should be close to 0 at observed points. Implicit Geometric Regularization (IGR) (Gropp et al., 2020) is a regularization based on the theory of Eikonal partial differential equations, which states that any function $F$ satisfying

$$F(x) = 0 \ \forall x \in B, \quad \|\nabla F(x)\| = 1 \ \forall x \in \mathbb{R}^3 \quad (4)$$

is a signed distance function for the surface $B$. To better encourage our decoder to describe a valid SDF, we therefore augment our loss function by an extra term $\mathcal{L}_{eik} = E[|\|\nabla SDF_\theta(x; h_k)\| - 1|]$. We estimate this expectation by sampling $x$ in the soft neighborhood of the object surface. For more details, we refer the reader to Section 4.

### 3.4 NORMAL VECTOR REGULARIZATION

If the training dataset contains ground-truth surface normal vectors, we can also add a regularization term to encourage the gradient of the estimated SDF at observed points to agree with the true normal vectors. This is a regularization used in the original IGR (Gropp et al., 2020) implementation as well.

### 3.5 LOSS FUNCTION AND POST-ENCODER LATENT OPTIMIZATION

The loss function we minimize at training time for object $k$ is given by

$$\mathcal{L} = \mathcal{L}_k(\theta) + \lambda \mathcal{L}_{eik}, \tag{5}$$

where $\lambda$ is an empirically chosen regularization parameter. At inference time, we can also further fine-tune the encoder output by optimizing the likelihood of the sparse observation. That is, if $\mu_0^k$ is the the mean of the encoder conditioned against $D_k$ (i.e. $E[h|D_k]$), we additionally apply the following iterative updates on the encoder's mean:

$$\mu_t^k \longleftarrow \mu_{t-1}^k + \sum_{(x_n^k, s_n^k) \in D_k} \nabla_h \log p(s_n^k | x_n^k, h) \Big|_{\mu_{t-1}^k}. \tag{6}$$

At inference time, we feed $\bar{\mu}_T^k$ to the decoder instead of $\mu_0^k$. When we apply this *post-encoder latent optimization*, our method becomes a version of IGR that is additionally equipped with an encoder.

## 4 EXPERIMENTS

We conducted a series of experiments to evaluate the efficacy of our proposed Meta-Shape Completion (MSC) method, relative to other well-known published methods. Particularly, we focus on very sparse scenarios, and show that the introduction of Bayesian meta-learning enables the generation of state-of-the-art shape predictions from a very small number of observed samples. Similarly, we show that our proposed method generalizes better to novel and unseen objects.

### 4.1 DATASETS

**ShapeNet (Chang et al., 2015).** We used the Synthetic ShapeNet CAD models as the primary source of evaluation for this paper. Following the procedure described in (Park et al., 2019), we first applied an affine transformation to each object, so the center of mass is located at the origin, and rescaled all points so that the maximum distance from vertex to origin is 1. To exclusively sample points on the object-surfaces, we first generated 100 view-points uniformly around each object, and used OpenGL to obtain images from these view-points. We then identified the set of points that are observable from these 100 view points as *surface points*, and took samples from these points. This modified dataset is henceforth referred to as *Disemboweled ShapeNet*. For IGR and MSC, we used the above procedure to generate $200k$ pairs of coordinate and normal vectors on each object surface. For DeepSDF, we followed the original procedure and computed the ground-truth SDF at randomly generated $500k$ points in addition to the $200k$ surface points. For OccNet, we randomly sampled $100k$ points from those used in DeepSDF, and annotated whether they are inside or outside the object. For PCN, we sampled $16384$ surface points as ground-truth.

**ICL-NUIM (Handa et al., 2014).** This is a dataset consisting of RGB-D images from two different scenes: living rooms and office rooms. The task on this dataset is to reconstruct 3D shapes from depth images. As pre-processing, we normalized location and scale of each scene using the same procedure described above for ShapeNet, and used Open3D (Zhou et al., 2018) to obtain normal vectors. Because Open3D sometimes produces normal vectors with inconsistent orientations, we conducted an extra step of inconsistency correction in which we made their inner products with the camera vector to be all positive.

### 4.2 EVALUATION

As our evaluation metric for each method, we calculated the Chamfer distance (Fan et al., 2017a) between ground-truth and generated point clouds. If $\mathcal{P}_1$ and $\mathcal{P}_2$ are two point clouds in 3D space, the

Table 1: Results on ShapeNet trained categories (Mean Chamfer Distance per point).

| Method | Raw Chamfer Distance (x1000) | | | | Normalized Chamfer Distance (w.r.t. MSC w/o opt) | | | |
|---|---|---|---|---|---|---|---|---|
| | 50 | 100 | 300 | 1000 | 50 | 100 | 300 | 1000 |
| Gauss Densification | 3.85 ±2.08 | 1.83 ±1.01 | 0.56 ±0.32 | 0.17 ±0.09 | 6.30 ±7.89 | 6.53 ±7.09 | 3.91 ±4.13 | 1.45 ±1.45 |
| PCN (Yuan et al., 2018) | 2.50 ±1.70 | 1.19 ±0.81 | 0.32 ±0.23 | **0.15 ±0.17** | 3.90 ±5.19 | 3.92 ±4.33 | 1.70 ±1.47 | **0.71 ±0.37** |
| OccNet (Mescheder et al., 2019) | 6.58 ±9.57 | 1.76 ±5.97 | 0.61 ±4.02 | 0.45 ±3.22 | 16.74 ±61.39 | 9.42 ±62.91 | 4.55 ±57.40 | 3.55 ±57.59 |
| DeepSDF (Park et al., 2019) | 5.38 ±3.51 | 5.06 ±3.44 | 4.72 ±3.31 | 4.58 ±3.32 | 13.16 ±25.95 | 24.62 ±42.41 | 37.89 ±54.77 | 43.50 ±60.37 |
| IGR (Gropp et al., 2020) | 3.61 ±4.54 | 3.33 ±4.36 | 3.14 ±4.24 | 3.09 ±4.21 | 8.64 ±23.38 | 15.65 ±37.04 | 24.28 ±51.47 | 28.32 ±58.50 |
| MSC (w/o opt) | 1.53 ±5.06 | 0.68 ±1.37 | 0.40 ±0.99 | 0.39 ±2.42 | 1 | 1 | 1 | 1 |
| MSC (w/ opt) | **0.67 ±0.80** | **0.38 ±0.92** | **0.29 ±0.94** | 0.29 ±1.29 | **0.82 ±1.92** | **0.90 ±10.93** | 1.03 ±7.07 | 1.24 ±13.47 |

Table 2: Results on ShapeNet novel categories (Mean Chamfer Distance per point).

| Method | Raw Chamfer Distance (x1000) | | | | Normalized Chamfer Distance (w.r.t. MSC w/o opt) | | | |
|---|---|---|---|---|---|---|---|---|
| | 50 | 100 | 300 | 1000 | 50 | 100 | 300 | 1000 |
| Gauss Densification | 2.97 ±1.65 | 1.42 ±0.80 | 0.43 ±0.26 | **0.13 ±0.08** | 4.33 ±4.98 | 4.02 ±4.73 | 2.09 ±2.58 | 0.75 ±0.92 |
| PCN (Yuan et al., 2018) | 2.06 ±1.44 | 0.99 ±0.70 | **0.34 ±0.32** | 0.22 ±0.28 | 2.94 ±3.98 | 2.74 ±3.88 | 1.18 ±1.20 | **0.60 ±0.32** |
| OccNet (Mescheder et al., 2019) | 12.30 ±18.08 | 5.08 ±12.85 | 1.37 ±6.02 | 0.68 ±3.80 | 22.44 ±53.69 | 12.76 ±39.53 | 3.55 ±17.52 | 1.53 ±8.58 |
| DeepSDF (Park et al., 2019) | 7.86 ±4.10 | 7.63 ±4.11 | 7.34 ±4.06 | 7.21 ±4.11 | 15.05 ±22.91 | 26.07 ±37.53 | 38.73 ±51.87 | 44.42 ±60.01 |
| IGR (Gropp et al., 2020) | 3.99 ±3.47 | 3.65 ±3.27 | 3.48 ±3.27 | 3.42 ±3.28 | 7.24 ±12.83 | 11.20 ±18.75 | 15.34 ±24.85 | 16.72 ±26.53 |
| MSC (w/o opt) | 1.60 ±5.13 | 0.90 ±2.86 | 0.62 ±1.74 | 0.53 ±1.20 | 1 | 1 | 1 | 1 |
| MSC (w/ opt) | **0.68 ±0.71** | **0.47 ±0.57** | 0.41 ±0.50 | 0.40 ±0.50 | **0.75 ±0.85** | **0.79 ±0.80** | **0.98 ±1.22** | 1.12 ±1.29 |

Chamfer distance between them is given by:

$$d(\mathcal{P}_1, \mathcal{P}_2) = \sum_{i \in \mathcal{P}_1} \min_{j \in \mathcal{P}_2} |x_i - x_j|_2^2 / |\mathcal{P}_1| + \sum_{j \in \mathcal{P}_2} \min_{i \in \mathcal{P}_1} |x_i - x_j|_2^2 / |\mathcal{P}_2| \tag{7}$$

Let $\mathcal{P}_k^{gt}$ be the ground-truth point cloud of the object $k$, and $\hat{\mathcal{P}}_k^M$ be the point cloud of object $k$ generated by method $\mathcal{M}$. To evaluate the performance of method $\mathcal{M}$, we computed $d_k^{\mathcal{M}} := d(\mathcal{P}_k^{gt}, \hat{\mathcal{P}}_k^{\mathcal{M}})$ for each $k$ and used it to calculate the following values:

$$\bar{d}_{ave}^{\mathcal{M}} = \frac{1}{K} \sum_{k=1}^{K} d_k^{\mathcal{M}} \quad , \quad \bar{d}_{norm}^{\mathcal{M}} = \frac{1}{K} \sum_{k=1}^{K} (d_k^{\mathcal{M}} / d_k^{\mathcal{M}_0}) \tag{8}$$

where $\mathcal{M}_0$ is our proposed method, so that $\bar{d}_{norm}^M$ is a measure of method $M$'s performance relative to our own performance. We refer to the former value as *average chamfer distance*, and the latter as *normalized chamfer distance*. For the ICL-NUIM experiments we used an *assymetric* variation of the Chamfer Distance, where only the first sum term in Equation 7 is used. This is because the ground-truth point clouds for this dataset are very sparse and lack many portions of the scene to be reconstructed.

To generate ground-truth point clouds, we randomly sampled $30k$ points from the raw meshes in our *Disemboweled Shapenet* dataset. Each point on the object surface was sampled in two steps: first, we selected a triangle on the polygonal mesh with probability proportional to the area of the triangle. Then, we sampled a point from the uniform distribution over the selected triangle. DeepSDF, IGR and OccNet use implicit functions to represent the geometry of each object, so their decoders do not directly output a point cloud. For these methods, we used Marching Cubes (Lorensen & Cline, 1987) to generate a point cloud of size $30k$. PCN, on the other hand, directly outputs 3D points from each set of sparse observations. Following the original implementation, we used those to generate a point cloud of size 16384.

### 4.3 SHAPE COMPLETION ON SHAPENET

To verify the ability of our proposed method to complete the shape of unseen categories, we split the object-categories of *Disembowled ShapeNet* into two groups: one to be used to train the model (training), and one to be used as newly-encountered object type (novel). For the exact splits, please refer to the Appendix. In the first set of experiments, we trained the model on the training dataset of training categories, and evaluated the model on the test set of the training categories. Results for this first set of experiments can be found on Table 1. In the second set of experiments, we trained each model on the training set of training categories and evaluated on the model on the test set of novel categories. Results for this second set of experiments can be found on Table 2. In both cases, our proposed method (MSC) was evaluated with and without post-encoder latent optimization.

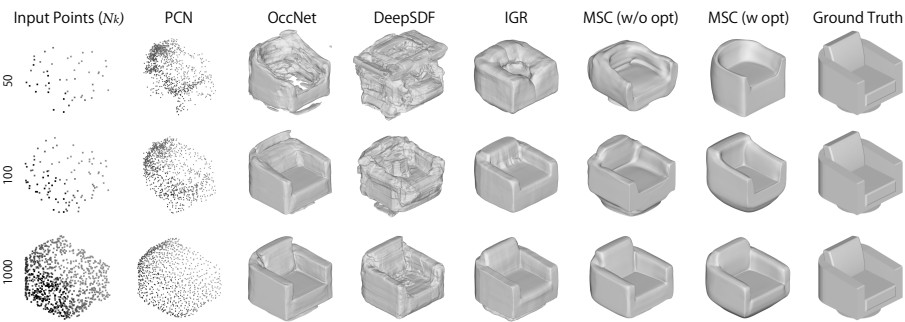

Figure 2: Shape completion results on ShapeNet for *chairs* (training category).

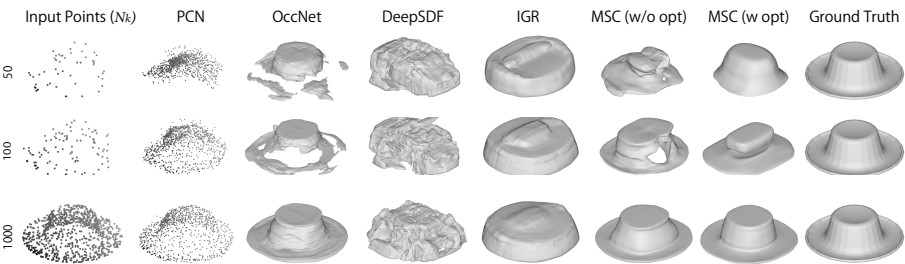

Figure 3: Shape completion results on ShapeNet for *caps* (novel category).

As we can see in both tables, our proposed method consistently outperforms all the other methods for both novel and training categories when the number of observation points is 100 or less, while achieving competitive results when higher densities are available. In particular, when the number of observations is 50, our method greatly outperforms PCN, the current SOTA in these tasks.

The strength of our method can be verified qualitatively as well. Figures 2 and 3 are examples of completed shapes obtained by different methods for various ShapeNet categories. As shown, our method tends to output smooth surface predictions regardless of the number of observations. This is most likely because our encoder is learning a task-agnostic property of *smoothness*, that can be transferred across different objects. Meanwhile, the performance of IGR differs drastically across trained and novel categories, both in terms of Chamfer distance and output appearance. DeepSDF performs poorly on all categories, most likely because it is not allowed to use ground-truth SDF at test-time. When the number of observations is small, DeepSDF tends to output a complex nonsensical surface that looks like a combination of all the training objects. It also tends to produce disconnected artifacts that are located far away from the real surface. OccNet also tends to produce many artifacts when the number of observations is small. We hypothesize that these methods are failing to isolate category-agnostic properties from category-specific properties. As a general remark, all methods struggle with reproducing detailed topology even with a large number of observations, particularly in the case of holes and large gaps, possibly due to model limitations. For a detailed analysis of the category-wise performance of various methods, please refer to the Appendix.

## 4.4 SCENE COMPLETION OF ICL-NUIM DATASET

Table 3 summarizes the performance of various methods on ICL-NUIM, for the task of scene completion. Similarly to ShapeNet, MSC significantly outperforms all others when the number of observations is 300 or less, while still achieving competitive results when a higher density of observations is available. Figures 4 and 5 contain examples of scene completion results produced by different methods. DeepSDF and OccNet both tend to produce very complex shapes with many disconnected artifacts, and DeepSDF in particular seems to not converge as we increase the number of observations. PCN performs poorly when compared to the shape completion task on ShapeNet, mostly densifying observed areas rather than extrapolating this information to other portions of the scene. Our method, on the other hand, succeeds in correctly recognizing the boundary of objects even when large areas, such as walls and the floor, are missing in the ground-truth.

Table 3: Results on the ICL-NUIM dataset (Mean Asymmetric Chamfer Distance per point).

| Method | Raw Asymmetric Chamfer Distance | | | | Normalized Asymmetric Chamfer Distance (w.r.t. MSC w/o opt) | | | |
|---|---|---|---|---|---|---|---|---|
| | 50 | 100 | 300 | 1000 | 50 | 100 | 300 | 1000 |
| Gauss Densification | 7.15 ±3.22 | 2.81 ±1.28 | 0.66 ±0.29 | 0.22 ±0.07 | 21.33 ±14.64 | 12.46 ±6.87 | 3.85 ±1.74 | 1.43 ±0.50 |
| PCN (Yuan et al., 2018) | 2.76 ±2.39 | 0.95 ±0.80 | 0.28 ±0.25 | 0.15 ±0.11 | 7.94 ±8.15 | 4.11 ±3.62 | 1.58 ±1.27 | 0.96 ±0.53 |
| OccNet (Mescheder et al., 2019) | 0.49 ±0.43 | 0.30 ±0.16 | 0.25 ±0.15 | 0.24 ±0.15 | 1.45 ±1.59 | 1.32 ±0.55 | 1.47 ±0.44 | 1.55 ±0.42 |
| DeepSDF (Park et al., 2019) | 3.33 ±5.10 | 2.12 ±5.48 | 1.03 ±2.71 | 0.57 ±1.36 | 10.09 ±20.04 | 9.51 ±34.35 | 5.91 ±17.61 | 3.45 ±8.17 |
| IGR (Gropp et al., 2020) | 3.11 ±5.76 | 2.28 ±5.01 | 1.62 ±4.28 | 1.30 ±4.07 | 8.49 ±17.77 | 9.02 ±19.90 | 8.77 ±22.49 | 7.98 ±24.74 |
| MSC (w/o opt) | 0.62 ±1.37 | 0.32 ±0.74 | 0.21 ±0.56 | 0.19 ±0.53 | 1 | 1 | 1 | 1 |
| MSC (w/ opt) | **0.20 ±0.33** | **0.15 ±0.15** | **0.14 ±0.03** | **0.14 ±0.03** | **0.59 ±1.36** | **0.66 ±0.57** | **0.80 ±0.16** | **0.87 ±0.14** |

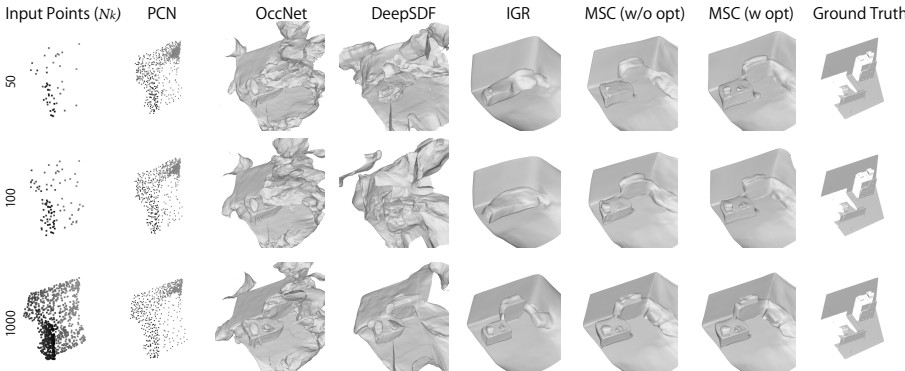

Figure 4: Scene completion results on ICL-NUIM for *office rooms*.

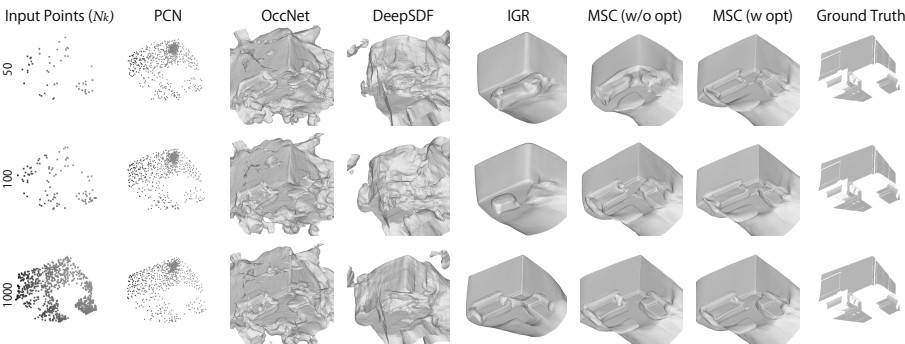

Figure 5: Scene completion results on ICL-NUIM for *living rooms*.

## 5 CONCLUSION

In this paper we introduce the concept of meta-learning to the task of shape completion using implicit representations of 3D surfaces. Our proposed encoder mechanism allows the learning of object-agnostic properties separately from object-specific properties, thus succeeding in training a model that can consistently produce smooth predictions from highly sparse observations, achieving state-of-the-art under these conditions. Although we have used IGR, an SDF-based method, as the basis for our implementation, our proposed meta-shape completion algorithm (MSC) can be equally applied to different implicit surface representations by simply changing the decoder. Furthermore, in this paper we have used a simple MLP-based encoder to learn task-specific parameters, while other methods like PCN use hierarchical models to capture both global and local geometric properties of objects. We believe that using these more complex models will lead to substantial improvements to our proposed method, however this is left for future work. In conclusion, there seems to be a lot of room left for the application of meta-learning to scene completion tasks, and further studies in this direction may allow us to develop models that can be used in a wide range of applications.

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

## A APPENDIX

### A.1 IMPLEMENTATION DETAILS

**DeepSDF (Park et al., 2019) and IGR (Gropp et al., 2020)**. We implemented both methods using Pytorch (Paszke et al., 2017), matching their model architectures, initialization procedures and published results. As done in the original study of (Park et al., 2019), however, we used ReLu in place of *softplus* for the Deep SDF decoder. To create the mesh with which the Chamfer distance is calculated, we used the Marching Cubes algorithm (Lorensen & Cline, 1987) with $256^3$ resolution to convert SDF to mesh. To find the optimal $z$ with respect to the likelihood, we applied gradient descent from a random initial value $z_0$ sampled from $N(0, 0.01^2)$. For the stepsize, we chose $lr * error_0$, where $error_0$ is the error computed with $z_0$. We used the Adam optimizer, with batch size $b = 32$, alpha $\alpha = 3.2 \times 10^{-4}$, and learning rates $lr = 1.0 \times 10^{-3}$ for DeepSDF and $lr = 1.0 \times 10^{-4}$ for IGR. We trained for 5000 epochs, halving the learning rate at every 500 epochs. We used 16384 surface points to compute the loss for each object. To generate points for the evaluation of the Eikonal term, we randomly selected $5k$ points from the 16384 on the surface, and sampled one point each from a Gaussian distribution centered around it, with variance of $4.0 \times 10^{-2}$ and $\lambda = 1.0$ as the Eikonal regularization parameter. We trained our IGR model with the version of the algorithm in (Gropp et al., 2020) that uses the surface normal vectors.

**OccNet (Mescheder et al., 2019)**. We used the authors' official Pytorch implementation[1] and trained under the same conditions as described in the original paper. For the encoder architecture (Figure 6), we used the *ResNetPointNet* class, and for the decoder architecture we used the *DecoderBatchNorm* class. We trained the decoder over $5k$ epochs with batch size $b = 64$ and learning rate $lr = 1.0 \times 10^{-4}$. For each object, we used 300 surface points to the encoder, and produced 2048 binary points from the decoder. To stabilize training, we followed the same procedure described in the original paper and added a Gaussian noise with standard deviation $\sigma = 5.0 \times 10^{-3}$ to the encoder input.

**PCN (Yuan et al., 2018)**. We implemented PCN using PyTorch based on the authors' official Tensorflow implementation[2], matching their model architectures, initialization procedures and published results. To train, we followed the same procedure as the original paper, feeding $k$ surface points to the encoder. We used batch size $b = 32$. Starting from the initial value of $0.0001$, we reduced the learning rate by a factor of $0.7$ at every $50k$ iterations and trained the model for a total of $5k$ epochs.

**MSC (Ours).** Figure 1 of the main paper illustrates our proposed encoder network. The dimension of $h$ (i.e., the output of the encoder) is set to 1024 for ShapeNet40 and 512 for the ICL-NUIM dataset. We used the same decoder as IGR. At training time, we set the number of contextual surface points to be 16384 and evaluated the loss using target sample points of size ranging from $1 \sim 5000$, chosen at random at each iteration. We trained our model for $5k$ epochs with batch size $b = 32$ and learning rate $lr = 10^{-4}$, using the same Eikonal regularization term as IGR. For the ShapeNet experiment, training the model required 16 days with 16 GPUs (NVIDIA V100). The encoding of 1000 contextual points took 0.003 seconds on average, and 22.660 seconds to create the mesh surface using the Marching

---

[1] https://github.com/autonomousvision/occupancy_networks
[2] https://github.com/wentaoyuan/pcn

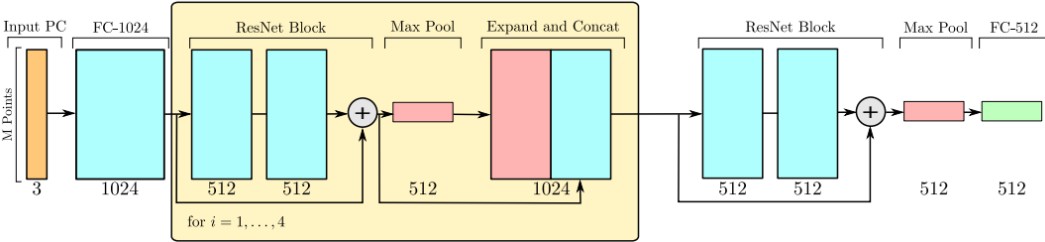

Figure 6: Encoder of OccNet (Mescheder et al., 2019). OccNet uses a version of PointNet that carries out max pooling procedures in multiple places. Compared to our encoder (Figure 1, main paper) this architecture is much more complex. Future work will involve extending MSC to different encoder architectures.

Cubes algorithm (which requires evaluation at multiple points). For the ICL-NUIM experiment, it took 30 hours with 32 GPUs to train the entire model.

## A.2 RESULTS ON SHAPENET

Tables 4 are 5 contain more detailed shape completion results for each ShapeNet training category, and Table 6 contains similar results for each ShapeNet novel category. Qualitative results for selected objects can be found in Figures 2-12. Generally speaking, performance of all methods vary greatly across categories, and as expected, all methods perform better on trained than novel categories. Some objects were particularly difficult for all methods, such as *headphones*, for which it is difficult to determine whether the target object is ring-shaped or not until there is a large amount of contextual points. Following the same logic, all methods also performed poorly on objects with intricate details, such as motorcycles and keyboards.

As described in the main paper, PCN tends to outperform MSC when there is a large number of observations. However, in many of theses cases simple Gaussian densification also yields good results. In fact, Gauss densification outperforms all methods on most categories when the number of observations is as large as 1000. This suggests that the representation power of these methods is still not sufficient. Meanwhile, MSC outperforms all methods on most of both novel and training categories when the number of observations is small. Particularly, when the number of observations is 50, our method with post-encoder optimization outperforms all methods on all novel categories. When the number of observations is 100, MSC with post-encoder optimization outperforms other methods in 10 out of 15 categories. However, we note that the categories where MSC is outperformed (*pistol*, *motorcycle*, *guitar*, *earphone*, *microphone*) correspond to objects with fine details, and in 2 of them the best performing method is Gauss densification.

IGR, which represents a version of our method without the encoder, sometimes produced nonsensical outputs even with a large number of observations, apparently switching between different objects. Similarly, DeepSDF has a tendency of producing outputs that look like a combination of many objects. It is possible that, because IGR and DeepSDF are not equipped with an encoder, they are indeliberately mixing object-specific with object-agnostic features by encoding some object-specific properties into their decoder model. DeepSDF also produced disconnected artifacts in many cases, mostly because we only allow the use of contextual observations at test time (e.g. information like *ground truth sdf values at points not on the surface* and *ground truth normal vectors* are inaccessible at test time with sparse observations). OccNet also produced such artifacts when the number of observations is very small. Our proposed MSC method, on the other hand, makes relatively conservative smooth prediction when the observations are too sparse, and incrementally increases the complexity of the output shapes as more points become available.

## A.3 RESULTS ON ICL-NUIM

Figures 13-15 are selected scene completion results obtained on the ICL-NUIM dataset. Although both the ground-truth and contextual point-cloud have missing large portions because of data collection procedure (collected from a single view), our MSC is able to generate smooth surfaces that cover the unobserved areas of the environment. Interestingly, in this case IGR also does a good job producing smooth surfaces, most likely because all rooms are somewhat familiar to each other (i.e. object-specific properties can be shared between classes without significant detrimental effects). DeepSDF again produces disconnected artifacts on this dataset as well, Failures cases for DeepSDF indicates that the Eikonal term plays an important role in producing smooth surfaces, especially when the model is not allowed to use normal vectors. Finally, PCN fails to reconstruct unobserved areas of the environment, producing instead what looks like a densified version of observed points.

Table 4: Results for each ShapeNet trained category (1/2)

| Category | Context | Method | | | | | | |
|---|---|---|---|---|---|---|---|---|
| | | Gauss densification | PCN | OccNet | DeepSDF | IGR | MSC (w/o opt) | MSC (w/ opt) |
| Plane | 50 | 1.22 ± 0.46 | 0.84 ± 0.44 | 1.45 ± 2.84 | 3.33 ± 1.56 | 3.07 ± 1.62 | 0.69 ± 1.60 | **0.31 ± 0.40** |
| | 100 | 0.54 ± 0.19 | 0.36 ± 0.23 | 0.37 ± 1.07 | 3.34 ± 1.54 | 2.84 ± 1.43 | 0.40 ± 1.49 | **0.19 ± 0.33** |
| | 300 | 0.16 ± 0.06 | **0.12 ± 0.12** | 0.14 ± 0.66 | 3.32 ± 1.72 | 2.71 ± 1.34 | 0.23 ± 0.53 | 0.17 ± 0.32 |
| | 1000 | **0.05 ± 0.02** | 0.07 ± 0.09 | 0.10 ± 0.47 | 3.38 ± 1.71 | 2.70 ± 1.26 | 0.20 ± 0.42 | 0.18 ± 0.34 |
| Garbage Can | 50 | 6.30 ± 1.24 | 4.41 ± 1.47 | 3.84 ± 3.74 | 1.93 ± 1.00 | **0.94 ± 0.93** | 1.96 ± 1.34 | 1.01 ± 0.67 |
| | 100 | 3.03 ± 0.56 | 1.97 ± 0.56 | 0.83 ± 1.01 | 1.64 ± 0.90 | 0.80 ± 0.82 | 0.79 ± 0.58 | **0.56 ± 0.45** |
| | 300 | 0.97 ± 0.18 | 0.44 ± 0.17 | 0.53 ± 1.09 | 1.39 ± 0.92 | 0.74 ± 0.85 | 0.42 ± 0.34 | **0.37 ± 0.31** |
| | 1000 | 0.29 ± 0.06 | **0.18 ± 0.12** | 0.45 ± 0.93 | 1.22 ± 0.82 | 0.70 ± 0.75 | 0.35 ± 0.28 | 0.32 ± 0.29 |
| Basket | 50 | 6.28 ± 1.75 | 4.14 ± 1.49 | 5.61 ± 3.68 | 3.26 ± 1.36 | 2.43 ± 1.26 | 1.69 ± 1.11 | **0.88 ± 0.94** |
| | 100 | 3.05 ± 0.94 | 2.09 ± 0.72 | 2.25 ± 2.12 | 3.15 ± 1.34 | 2.26 ± 1.20 | 1.01 ± 1.12 | **0.58 ± 0.89** |
| | 300 | 0.93 ± 0.29 | **0.46 ± 0.41** | 1.46 ± 2.05 | 3.02 ± 1.34 | 2.12 ± 1.23 | 0.55 ± 0.81 | 0.47 ± 0.73 |
| | 1000 | 0.28 ± 0.11 | **0.27 ± 0.37** | 1.11 ± 1.54 | 2.76 ± 1.04 | 2.15 ± 1.15 | 0.59 ± 0.91 | 0.47 ± 0.69 |
| Bathtub | 50 | 4.96 ± 1.64 | 3.07 ± 1.42 | 8.63 ± 6.15 | 4.20 ± 1.53 | 1.87 ± 1.06 | 1.39 ± 1.06 | **0.64 ± 0.51** |
| | 100 | 2.44 ± 0.89 | 1.40 ± 0.65 | 3.41 ± 4.37 | 4.16 ± 1.73 | 1.73 ± 1.18 | 0.63 ± 0.68 | **0.36 ± 0.39** |
| | 300 | 0.78 ± 0.28 | 0.36 ± 0.16 | 0.78 ± 1.47 | 4.15 ± 1.67 | 1.64 ± 1.11 | 0.38 ± 0.49 | **0.27 ± 0.35** |
| | 1000 | 0.23 ± 0.08 | **0.16 ± 0.12** | 0.54 ± 0.90 | 3.86 ± 1.57 | 1.65 ± 1.05 | 0.34 ± 0.43 | 0.30 ± 0.40 |
| Bottle | 50 | 3.95 ± 1.29 | 3.33 ± 1.55 | 7.05 ± 9.35 | 3.17 ± 1.79 | **0.53 ± 1.47** | 1.23 ± 1.08 | 0.57 ± 0.63 |
| | 100 | 1.86 ± 0.67 | 1.45 ± 0.61 | 0.32 ± 0.44 | 2.64 ± 1.63 | 0.50 ± 1.41 | 0.42 ± 0.48 | **0.23 ± 0.39** |
| | 300 | 0.56 ± 0.20 | 0.29 ± 0.16 | 0.16 ± 0.36 | 1.89 ± 1.18 | 0.43 ± 1.39 | 0.22 ± 0.31 | **0.16 ± 0.33** |
| | 1000 | 0.16 ± 0.06 | **0.09 ± 0.08** | 0.12 ± 0.24 | 1.61 ± 1.09 | 0.41 ± 1.31 | 0.18 ± 0.28 | 0.13 ± 0.27 |
| Bowl | 50 | 3.78 ± 1.02 | 2.95 ± 1.33 | 5.68 ± 3.19 | 4.22 ± 1.68 | 2.41 ± 1.09 | 0.58 ± 0.63 | **0.21 ± 0.24** |
| | 100 | 1.87 ± 0.51 | 1.05 ± 0.39 | 3.26 ± 3.70 | 3.82 ± 1.54 | 2.11 ± 1.10 | 0.21 ± 0.16 | **0.08 ± 0.05** |
| | 300 | 0.59 ± 0.19 | 0.26 ± 0.15 | 2.45 ± 4.00 | 3.93 ± 1.51 | 1.89 ± 1.02 | 0.12 ± 0.05 | **0.07 ± 0.05** |
| | 1000 | 0.19 ± 0.06 | 0.11 ± 0.05 | 1.68 ± 3.30 | 3.48 ± 1.42 | 1.85 ± 1.08 | 0.11 ± 0.05 | **0.07 ± 0.05** |
| Cabinet | 50 | 6.31 ± 1.46 | 3.79 ± 1.46 | 10.59 ± 6.63 | 3.30 ± 1.77 | **0.72 ± 0.88** | 1.58 ± 1.00 | 0.93 ± 0.56 |
| | 100 | 3.10 ± 0.67 | 1.66 ± 0.63 | 2.27 ± 3.27 | 2.47 ± 1.53 | 0.68 ± 0.99 | 0.71 ± 0.47 | **0.51 ± 0.31** |
| | 300 | 0.98 ± 0.20 | 0.35 ± 0.14 | **0.21 ± 0.32** | 2.05 ± 1.39 | 0.60 ± 0.90 | 0.28 ± 0.26 | 0.31 ± 0.30 |
| | 1000 | 0.28 ± 0.06 | **0.12 ± 0.08** | 0.18 ± 0.26 | 1.80 ± 1.21 | 0.61 ± 0.88 | 0.22 ± 0.24 | 0.30 ± 0.28 |
| Camera | 50 | 4.18 ± 1.39 | 4.01 ± 1.91 | 2.61 ± 2.38 | 4.67 ± 2.95 | 3.71 ± 2.58 | 2.61 ± 2.62 | **1.20 ± 0.88** |
| | 100 | 2.18 ± 0.82 | 2.14 ± 1.23 | 0.96 ± 0.85 | 4.25 ± 2.98 | 3.42 ± 2.41 | 1.64 ± 1.50 | **0.91 ± 0.79** |
| | 300 | 0.66 ± 0.22 | 0.75 ± 0.54 | **0.43 ± 0.53** | 3.91 ± 2.95 | 3.45 ± 2.50 | 1.06 ± 1.09 | 0.73 ± 0.76 |
| | 1000 | **0.19 ± 0.06** | 0.48 ± 0.49 | 0.37 ± 0.33 | 3.91 ± 2.65 | 3.09 ± 2.36 | 1.19 ± 1.67 | 0.81 ± 0.72 |
| Can | 50 | 6.70 ± 1.38 | 4.47 ± 1.14 | 3.01 ± 1.62 | 1.85 ± 0.53 | **0.16 ± 0.18** | 1.95 ± 0.94 | 0.93 ± 0.63 |
| | 100 | 3.17 ± 0.54 | 2.32 ± 0.91 | 0.25 ± 0.27 | 1.49 ± 0.48 | **0.11 ± 0.10** | 0.56 ± 0.58 | 0.64 ± 0.78 |
| | 300 | 0.98 ± 0.15 | 0.37 ± 0.10 | 0.17 ± 0.30 | 1.34 ± 0.41 | 0.11 ± 0.12 | 0.13 ± 0.11 | **0.10 ± 0.15** |
| | 1000 | 0.28 ± 0.04 | 0.09 ± 0.04 | **0.05 ± 0.03** | 1.26 ± 0.40 | 0.10 ± 0.08 | 0.09 ± 0.08 | 0.10 ± 0.15 |
| Car | 50 | 3.97 ± 0.77 | 3.65 ± 1.35 | 2.16 ± 2.62 | 3.87 ± 1.30 | 0.79 ± 0.72 | 0.74 ± 0.52 | **0.50 ± 0.36** |
| | 100 | 1.95 ± 0.36 | 1.83 ± 0.55 | 0.37 ± 0.73 | 3.84 ± 1.15 | 0.61 ± 0.64 | 0.36 ± 0.27 | **0.27 ± 0.24** |
| | 300 | 0.62 ± 0.11 | 0.35 ± 0.10 | **0.16 ± 0.39** | 3.71 ± 1.11 | 0.46 ± 0.54 | 0.23 ± 0.25 | 0.19 ± 0.21 |
| | 1000 | 0.19 ± 0.03 | **0.11 ± 0.07** | 0.15 ± 0.33 | 3.56 ± 1.11 | 0.42 ± 0.56 | 0.19 ± 0.22 | 0.18 ± 0.20 |
| Cellular Phone | 50 | 3.44 ± 0.86 | 1.73 ± 0.81 | 20.11 ± 10.76 | 5.85 ± 3.19 | 0.60 ± 2.26 | 0.41 ± 0.44 | **0.20 ± 0.28** |
| | 100 | 1.74 ± 0.40 | 0.91 ± 0.37 | 4.08 ± 7.49 | 4.53 ± 3.18 | 0.58 ± 2.14 | 0.26 ± 1.21 | **0.09 ± 0.13** |
| | 300 | 0.58 ± 0.12 | 0.18 ± 0.07 | 0.34 ± 3.11 | 3.18 ± 2.67 | 0.58 ± 2.31 | 0.09 ± 0.18 | **0.07 ± 0.11** |
| | 1000 | 0.17 ± 0.03 | **0.05 ± 0.03** | 0.27 ± 2.80 | 2.67 ± 2.49 | 0.56 ± 2.30 | 0.08 ± 0.13 | 0.06 ± 0.08 |
| Chair | 50 | 4.51 ± 1.61 | 2.68 ± 1.40 | 8.52 ± 7.35 | 4.60 ± 1.62 | 2.69 ± 1.76 | 2.36 ± 9.64 | **0.83 ± 0.72** |
| | 100 | 2.08 ± 0.82 | 1.28 ± 0.71 | 1.58 ± 3.66 | 4.33 ± 1.62 | 2.38 ± 1.53 | 0.77 ± 0.91 | **0.35 ± 0.35** |
| | 300 | 0.62 ± 0.28 | 0.38 ± 0.20 | 0.35 ± 1.45 | 3.94 ± 1.52 | 2.23 ± 1.51 | 0.40 ± 0.43 | **0.23 ± 0.28** |
| | 1000 | 0.18 ± 0.08 | **0.16 ± 0.12** | 0.26 ± 1.04 | 3.74 ± 1.52 | 2.16 ± 1.41 | 0.33 ± 0.49 | 0.22 ± 0.30 |
| Clock | 50 | 3.77 ± 1.71 | 2.35 ± 1.44 | 9.84 ± 13.75 | 4.08 ± 3.22 | 1.29 ± 1.86 | 1.08 ± 1.17 | **0.58 ± 0.64** |
| | 100 | 1.88 ± 0.80 | 1.19 ± 0.84 | 2.47 ± 6.61 | 3.55 ± 3.14 | 1.18 ± 1.65 | 0.68 ± 1.29 | **0.41 ± 0.65** |
| | 300 | 0.60 ± 0.25 | 0.32 ± 0.22 | 0.68 ± 2.65 | 2.89 ± 3.22 | 1.18 ± 1.67 | 0.42 ± 0.83 | **0.30 ± 0.44** |
| | 1000 | 0.18 ± 0.08 | **0.16 ± 0.17** | 0.56 ± 2.53 | 2.44 ± 2.72 | 1.17 ± 1.77 | 0.37 ± 0.86 | 0.28 ± 0.36 |
| Dishwasher | 50 | 8.39 ± 1.09 | 5.95 ± 2.55 | 6.77 ± 2.26 | 2.52 ± 1.25 | **0.31 ± 0.37** | 2.48 ± 1.13 | 1.49 ± 0.80 |
| | 100 | 3.99 ± 0.55 | 3.01 ± 0.83 | 0.42 ± 0.62 | 1.96 ± 0.44 | **0.20 ± 0.32** | 1.26 ± 0.83 | 1.02 ± 0.63 |
| | 300 | 1.33 ± 0.16 | 0.41 ± 0.11 | **0.08 ± 0.16** | 1.98 ± 0.78 | 0.17 ± 0.30 | 0.25 ± 0.44 | 0.15 ± 0.31 |
| | 1000 | 0.37 ± 0.04 | 0.10 ± 0.05 | **0.10 ± 0.23** | 1.89 ± 0.67 | 0.17 ± 0.32 | 0.11 ± 0.12 | 0.12 ± 0.30 |
| Display | 50 | 3.74 ± 1.27 | 2.10 ± 1.51 | 6.42 ± 7.18 | 5.90 ± 2.81 | 2.34 ± 2.60 | 1.39 ± 3.25 | **0.49 ± 0.50** |
| | 100 | 1.88 ± 0.60 | 1.03 ± 0.67 | 1.72 ± 4.55 | 5.11 ± 2.44 | 2.17 ± 2.76 | 0.63 ± 1.28 | **0.29 ± 0.28** |
| | 300 | 0.62 ± 0.20 | 0.32 ± 0.18 | 0.53 ± 2.64 | 4.38 ± 2.25 | 2.14 ± 2.65 | 0.41 ± 0.67 | **0.26 ± 0.38** |
| | 1000 | 0.19 ± 0.05 | **0.14 ± 0.12** | 0.31 ± 1.96 | 4.01 ± 2.19 | 2.15 ± 2.39 | 0.37 ± 0.57 | 0.26 ± 0.40 |
| Faucet | 50 | 2.02 ± 1.24 | 1.68 ± 1.35 | 5.79 ± 13.81 | 10.42 ± 4.56 | 14.80 ± 8.31 | 2.10 ± 3.23 | **0.61 ± 0.93** |
| | 100 | 0.93 ± 0.60 | 0.80 ± 0.61 | 2.56 ± 10.08 | 10.15 ± 4.49 | 14.14 ± 7.89 | 1.16 ± 2.22 | **0.33 ± 0.52** |
| | 300 | **0.28 ± 0.18** | 0.37 ± 0.42 | 1.24 ± 7.29 | 9.48 ± 4.21 | 13.21 ± 7.44 | 0.92 ± 2.35 | 0.31 ± 0.62 |
| | 1000 | **0.09 ± 0.05** | 0.28 ± 0.40 | 0.57 ± 2.85 | 9.24 ± 4.21 | 13.19 ± 7.70 | 0.84 ± 2.26 | 0.28 ± 0.57 |
| File Cabinet | 50 | 7.18 ± 1.60 | 4.42 ± 1.56 | 8.69 ± 6.47 | 2.87 ± 1.75 | **0.92 ± 2.21** | 2.13 ± 1.01 | 1.14 ± 0.67 |
| | 100 | 3.59 ± 0.91 | 2.19 ± 0.96 | 1.46 ± 2.54 | 2.58 ± 1.74 | 0.88 ± 2.46 | 0.90 ± 0.76 | **0.71 ± 0.55** |
| | 300 | 1.07 ± 0.24 | 0.40 ± 0.26 | **0.15 ± 0.32** | 2.11 ± 1.43 | 0.78 ± 2.19 | 0.23 ± 0.31 | 0.33 ± 0.45 |
| | 1000 | 0.31 ± 0.07 | **0.14 ± 0.23** | 0.18 ± 0.39 | 1.99 ± 1.45 | 0.78 ± 2.24 | 0.17 ± 0.28 | 0.21 ± 0.36 |
| Helmet | 50 | 4.69 ± 0.83 | 3.66 ± 1.26 | 2.65 ± 1.70 | 1.94 ± 0.68 | 2.38 ± 1.61 | 2.38 ± 1.08 | **1.62 ± 0.87** |
| | 100 | 2.33 ± 0.37 | 1.72 ± 0.49 | 1.52 ± 0.85 | 1.83 ± 0.71 | 2.00 ± 1.29 | 1.79 ± 1.07 | **1.45 ± 1.21** |
| | 300 | 0.74 ± 0.15 | **0.53 ± 0.21** | 1.04 ± 0.89 | 1.71 ± 0.75 | 1.87 ± 1.07 | 1.48 ± 0.98 | 1.09 ± 1.02 |
| | 1000 | **0.21 ± 0.04** | 0.31 ± 0.19 | 0.98 ± 0.86 | 1.56 ± 0.58 | 1.87 ± 1.13 | 1.31 ± 0.96 | 1.04 ± 0.96 |
| Jar | 50 | 4.47 ± 1.51 | 3.62 ± 1.51 | 4.45 ± 6.73 | 2.52 ± 1.93 | 1.31 ± 1.33 | 2.05 ± 2.54 | **1.02 ± 0.84** |
| | 100 | 2.18 ± 0.72 | 1.77 ± 0.81 | 1.61 ± 2.50 | 2.22 ± 1.52 | 1.16 ± 1.18 | 1.08 ± 1.00 | **0.69 ± 0.77** |
| | 300 | 0.67 ± 0.24 | **0.49 ± 0.33** | 1.32 ± 3.35 | 1.78 ± 1.33 | 1.10 ± 1.28 | 0.85 ± 1.05 | 0.56 ± 0.77 |
| | 1000 | **0.20 ± 0.07** | 0.27 ± 0.31 | 1.47 ± 4.64 | 1.60 ± 1.07 | 1.08 ± 1.19 | 0.78 ± 0.94 | 0.55 ± 0.80 |
| Knife | 50 | 0.65 ± 0.32 | **0.41 ± 0.28** | 51.60 ± 27.64 | 12.54 ± 3.81 | 8.71 ± 4.06 | 0.93 ± 2.82 | 0.60 ± 2.26 |
| | 100 | 0.29 ± 0.15 | **0.18 ± 0.12** | 33.18 ± 31.98 | 12.40 ± 3.55 | 8.04 ± 3.49 | 0.49 ± 2.06 | 1.43 ± 8.07 |
| | 300 | 0.09 ± 0.04 | **0.07 ± 0.05** | 17.62 ± 29.70 | 11.06 ± 3.80 | 7.79 ± 3.61 | 0.16 ± 0.14 | 1.20 ± 7.11 |
| | 1000 | **0.04 ± 0.01** | 0.06 ± 0.04 | 11.11 ± 23.98 | 11.23 ± 3.64 | 7.84 ± 3.78 | 0.32 ± 1.37 | 1.98 ± 11.74 |

Table 5: Results for each ShapeNet trained category (2/2)

| Category | Context | Method | | | | | | |
|---|---|---|---|---|---|---|---|---|
| | | Gauss densification | PCN | OccNet | DeepSDF | IGR | MSC (w/o opt) | MSC (w/ opt) |
| Lamp | 50 | $2.45 \pm 1.65$ | $1.99 \pm 1.55$ | $9.29 \pm 15.61$ | $8.10 \pm 5.98$ | $6.15 \pm 7.96$ | $3.38 \pm 10.31$ | $\mathbf{1.00 \pm 1.35}$ |
| | 100 | $1.08 \pm 0.74$ | $1.02 \pm 0.84$ | $3.34 \pm 9.39$ | $7.69 \pm 6.16$ | $5.70 \pm 7.63$ | $1.78 \pm 3.22$ | $\mathbf{0.66 \pm 0.74}$ |
| | 300 | $\mathbf{0.31 \pm 0.24}$ | $0.42 \pm 0.41$ | $1.24 \pm 6.24$ | $7.09 \pm 5.93$ | $5.55 \pm 7.82$ | $1.22 \pm 1.96$ | $0.58 \pm 1.21$ |
| | 1000 | $\mathbf{0.09 \pm 0.06}$ | $0.31 \pm 0.35$ | $1.15 \pm 6.87$ | $6.89 \pm 5.88$ | $5.40 \pm 7.66$ | $1.52 \pm 7.89$ | $0.60 \pm 1.48$ |
| Laptop | 50 | $3.75 \pm 0.79$ | $1.86 \pm 0.82$ | $9.86 \pm 9.99$ | $10.40 \pm 2.83$ | $14.41 \pm 6.24$ | $2.09 \pm 8.32$ | $\mathbf{0.20 \pm 0.29}$ |
| | 100 | $1.73 \pm 0.34$ | $0.61 \pm 0.29$ | $1.87 \pm 4.93$ | $10.35 \pm 2.83$ | $13.52 \pm 6.08$ | $1.14 \pm 4.94$ | $\mathbf{0.10 \pm 0.20}$ |
| | 300 | $0.57 \pm 0.13$ | $0.18 \pm 0.07$ | $0.60 \pm 2.76$ | $10.22 \pm 2.47$ | $12.60 \pm 6.05$ | $0.14 \pm 0.34$ | $\mathbf{0.09 \pm 0.19}$ |
| | 1000 | $0.19 \pm 0.04$ | $\mathbf{0.08 \pm 0.04}$ | $0.42 \pm 2.35$ | $10.07 \pm 2.60$ | $12.77 \pm 5.41$ | $0.10 \pm 0.16$ | $0.09 \pm 0.21$ |
| Speaker | 50 | $6.31 \pm 2.22$ | $4.25 \pm 2.05$ | $8.28 \pm 7.44$ | $3.30 \pm 2.20$ | $\mathbf{1.06 \pm 1.41}$ | $1.81 \pm 1.10$ | $1.10 \pm 0.72$ |
| | 100 | $3.12 \pm 1.07$ | $2.03 \pm 0.88$ | $1.79 \pm 3.57$ | $2.78 \pm 1.94$ | $0.89 \pm 1.24$ | $0.83 \pm 0.65$ | $\mathbf{0.68 \pm 0.56}$ |
| | 300 | $0.98 \pm 0.34$ | $0.44 \pm 0.21$ | $\mathbf{0.31 \pm 0.80}$ | $2.45 \pm 1.95$ | $0.88 \pm 1.31$ | $0.41 \pm 0.45$ | $0.50 \pm 0.50$ |
| | 1000 | $0.29 \pm 0.10$ | $\mathbf{0.18 \pm 0.16}$ | $0.29 \pm 0.62$ | $2.23 \pm 1.81$ | $0.87 \pm 1.34$ | $0.34 \pm 0.42$ | $0.47 \pm 0.55$ |
| Mailbox | 50 | $3.41 \pm 2.33$ | $2.64 \pm 2.01$ | $8.66 \pm 12.04$ | $6.39 \pm 4.55$ | $4.52 \pm 3.61$ | $1.30 \pm 0.94$ | $\mathbf{0.60 \pm 0.46}$ |
| | 100 | $1.68 \pm 1.10$ | $1.32 \pm 0.97$ | $0.81 \pm 1.12$ | $5.92 \pm 4.28$ | $4.04 \pm 3.58$ | $0.76 \pm 0.61$ | $\mathbf{0.45 \pm 0.46}$ |
| | 300 | $0.52 \pm 0.36$ | $0.31 \pm 0.16$ | $\mathbf{0.15 \pm 0.18}$ | $5.14 \pm 3.81$ | $4.00 \pm 3.58$ | $0.44 \pm 0.50$ | $0.39 \pm 0.41$ |
| | 1000 | $0.16 \pm 0.10$ | $0.17 \pm 0.13$ | $\mathbf{0.14 \pm 0.17}$ | $5.05 \pm 3.87$ | $4.10 \pm 4.15$ | $0.46 \pm 0.60$ | $0.29 \pm 0.33$ |
| Microwave | 50 | $8.02 \pm 1.14$ | $5.70 \pm 1.35$ | $4.81 \pm 2.10$ | $3.21 \pm 0.94$ | $\mathbf{0.45 \pm 0.80}$ | $2.14 \pm 0.84$ | $1.56 \pm 1.12$ |
| | 100 | $3.80 \pm 0.37$ | $2.63 \pm 0.78$ | $0.63 \pm 0.61$ | $2.81 \pm 0.77$ | $\mathbf{0.33 \pm 0.64}$ | $0.95 \pm 0.71$ | $0.62 \pm 0.50$ |
| | 300 | $1.23 \pm 0.13$ | $0.39 \pm 0.16$ | $\mathbf{0.08 \pm 0.14}$ | $2.56 \pm 0.69$ | $0.33 \pm 0.77$ | $0.37 \pm 0.57$ | $0.31 \pm 0.37$ |
| | 1000 | $0.35 \pm 0.03$ | $0.10 \pm 0.04$ | $\mathbf{0.06 \pm 0.06}$ | $2.40 \pm 0.67$ | $0.34 \pm 0.76$ | $0.11 \pm 0.08$ | $0.32 \pm 0.35$ |
| Mug | 50 | $5.46 \pm 1.02$ | $4.58 \pm 1.82$ | $2.09 \pm 1.46$ | $2.16 \pm 0.52$ | $2.71 \pm 1.06$ | $1.53 \pm 0.96$ | $\mathbf{0.65 \pm 0.69}$ |
| | 100 | $2.73 \pm 0.41$ | $1.87 \pm 0.65$ | $0.66 \pm 0.84$ | $1.98 \pm 0.48$ | $2.35 \pm 0.89$ | $0.92 \pm 0.70$ | $\mathbf{0.32 \pm 0.39}$ |
| | 300 | $0.86 \pm 0.13$ | $0.42 \pm 0.22$ | $0.23 \pm 0.47$ | $1.80 \pm 0.46$ | $2.13 \pm 0.95$ | $0.39 \pm 0.39$ | $\mathbf{0.13 \pm 0.14}$ |
| | 1000 | $0.26 \pm 0.04$ | $0.18 \pm 0.14$ | $0.17 \pm 0.30$ | $1.72 \pm 0.44$ | $1.95 \pm 0.93$ | $0.29 \pm 0.29$ | $\mathbf{0.12 \pm 0.16}$ |
| Piano | 50 | $4.78 \pm 1.80$ | $3.34 \pm 1.76$ | $4.96 \pm 4.91$ | $5.13 \pm 2.85$ | $3.55 \pm 2.46$ | $1.66 \pm 1.11$ | $\mathbf{0.91 \pm 0.59}$ |
| | 100 | $2.33 \pm 0.91$ | $1.66 \pm 0.73$ | $0.68 \pm 0.61$ | $5.08 \pm 2.71$ | $3.34 \pm 2.43$ | $0.86 \pm 0.56$ | $\mathbf{0.57 \pm 0.37}$ |
| | 300 | $0.71 \pm 0.28$ | $0.47 \pm 0.24$ | $\mathbf{0.20 \pm 0.16}$ | $4.79 \pm 2.91$ | $3.34 \pm 2.33$ | $0.63 \pm 0.40$ | $0.42 \pm 0.30$ |
| | 1000 | $\mathbf{0.21 \pm 0.08}$ | $0.24 \pm 0.14$ | $0.22 \pm 0.17$ | $5.21 \pm 3.30$ | $3.38 \pm 2.53$ | $0.54 \pm 0.36$ | $0.39 \pm 0.32$ |
| Pillow | 50 | $4.46 \pm 0.89$ | $3.79 \pm 1.84$ | $3.07 \pm 2.21$ | $3.57 \pm 1.97$ | $0.86 \pm 0.90$ | $1.27 \pm 0.96$ | $\mathbf{0.65 \pm 0.73}$ |
| | 100 | $2.20 \pm 0.50$ | $2.14 \pm 0.86$ | $0.72 \pm 0.91$ | $3.09 \pm 1.56$ | $0.72 \pm 0.85$ | $0.31 \pm 0.23$ | $\mathbf{0.15 \pm 0.24}$ |
| | 300 | $0.67 \pm 0.14$ | $0.42 \pm 0.12$ | $\mathbf{0.07 \pm 0.11}$ | $3.10 \pm 2.05$ | $0.68 \pm 0.82$ | $0.22 \pm 0.26$ | $0.16 \pm 0.30$ |
| | 1000 | $0.19 \pm 0.03$ | $0.11 \pm 0.06$ | $\mathbf{0.06 \pm 0.08}$ | $2.84 \pm 1.56$ | $0.63 \pm 0.80$ | $0.18 \pm 0.22$ | $0.11 \pm 0.22$ |
| Pot | 50 | $5.04 \pm 1.86$ | $3.39 \pm 1.36$ | $5.50 \pm 6.50$ | $2.63 \pm 1.65$ | $1.97 \pm 1.37$ | $2.00 \pm 1.53$ | $\mathbf{0.95 \pm 0.71}$ |
| | 100 | $2.43 \pm 0.88$ | $1.68 \pm 0.72$ | $1.82 \pm 2.66$ | $2.37 \pm 1.60$ | $1.72 \pm 1.32$ | $1.26 \pm 1.12$ | $\mathbf{0.69 \pm 0.61}$ |
| | 300 | $0.78 \pm 0.27$ | $\mathbf{0.56 \pm 0.33}$ | $1.38 \pm 2.69$ | $2.07 \pm 1.36$ | $1.63 \pm 1.29$ | $0.87 \pm 0.84$ | $0.59 \pm 0.56$ |
| | 1000 | $\mathbf{0.24 \pm 0.09}$ | $0.35 \pm 0.30$ | $1.50 \pm 3.21$ | $1.99 \pm 1.32$ | $1.56 \pm 1.26$ | $0.89 \pm 0.98$ | $0.61 \pm 0.62$ |
| Printer | 50 | $6.01 \pm 1.68$ | $4.72 \pm 2.02$ | $3.29 \pm 2.77$ | $3.46 \pm 1.45$ | $1.86 \pm 1.27$ | $1.89 \pm 0.92$ | $\mathbf{1.35 \pm 0.90}$ |
| | 100 | $2.97 \pm 0.87$ | $2.39 \pm 1.00$ | $\mathbf{0.69 \pm 0.47}$ | $3.51 \pm 1.54$ | $1.61 \pm 1.24$ | $1.29 \pm 1.03$ | $0.79 \pm 0.53$ |
| | 300 | $0.94 \pm 0.26$ | $0.54 \pm 0.18$ | $\mathbf{0.24 \pm 0.26}$ | $3.01 \pm 1.38$ | $1.43 \pm 1.19$ | $0.67 \pm 0.55$ | $0.52 \pm 0.39$ |
| | 1000 | $0.27 \pm 0.07$ | $\mathbf{0.22 \pm 0.10}$ | $0.24 \pm 0.23$ | $3.08 \pm 1.28$ | $1.50 \pm 1.21$ | $0.48 \pm 0.36$ | $0.58 \pm 0.45$ |
| Rifle | 50 | $0.91 \pm 0.49$ | $0.60 \pm 0.41$ | $0.91 \pm 3.62$ | $8.07 \pm 2.66$ | $4.28 \pm 3.98$ | $0.55 \pm 0.64$ | $\mathbf{0.31 \pm 0.33}$ |
| | 100 | $0.40 \pm 0.22$ | $0.29 \pm 0.19$ | $\mathbf{0.11 \pm 0.15}$ | $7.74 \pm 2.56$ | $3.43 \pm 3.50$ | $0.33 \pm 0.37$ | $0.20 \pm 0.24$ |
| | 300 | $0.12 \pm 0.07$ | $0.10 \pm 0.07$ | $\mathbf{0.07 \pm 0.05}$ | $7.74 \pm 2.37$ | $2.70 \pm 2.97$ | $0.24 \pm 0.28$ | $0.17 \pm 0.31$ |
| | 1000 | $\mathbf{0.05 \pm 0.02}$ | $0.07 \pm 0.06$ | $0.08 \pm 0.05$ | $7.73 \pm 2.51$ | $2.43 \pm 2.90$ | $0.22 \pm 0.28$ | $0.18 \pm 0.30$ |
| Rocket | 50 | $1.14 \pm 0.61$ | $1.19 \pm 0.76$ | $0.41 \pm 0.19$ | $6.77 \pm 2.50$ | $4.82 \pm 4.61$ | $0.54 \pm 0.36$ | $\mathbf{0.29 \pm 0.25}$ |
| | 100 | $0.52 \pm 0.23$ | $0.59 \pm 0.32$ | $\mathbf{0.13 \pm 0.10}$ | $6.97 \pm 1.85$ | $4.75 \pm 3.76$ | $0.34 \pm 0.20$ | $0.25 \pm 0.31$ |
| | 300 | $0.16 \pm 0.07$ | $0.18 \pm 0.09$ | $\mathbf{0.10 \pm 0.10}$ | $7.72 \pm 3.12$ | $3.36 \pm 2.86$ | $0.29 \pm 0.22$ | $0.25 \pm 0.28$ |
| | 1000 | $\mathbf{0.05 \pm 0.02}$ | $0.09 \pm 0.07$ | $0.08 \pm 0.06$ | $8.11 \pm 2.79$ | $3.30 \pm 3.07$ | $0.25 \pm 0.19$ | $0.22 \pm 0.27$ |
| Sofa | 50 | $4.87 \pm 1.29$ | $2.87 \pm 1.29$ | $3.53 \pm 3.29$ | $5.46 \pm 1.91$ | $2.74 \pm 3.68$ | $1.22 \pm 0.84$ | $\mathbf{0.59 \pm 0.45}$ |
| | 100 | $2.45 \pm 0.64$ | $1.58 \pm 0.59$ | $0.44 \pm 1.11$ | $5.40 \pm 2.04$ | $2.57 \pm 3.63$ | $0.49 \pm 0.35$ | $\mathbf{0.26 \pm 0.25}$ |
| | 300 | $0.79 \pm 0.20$ | $0.41 \pm 0.15$ | $\mathbf{0.17 \pm 0.95}$ | $5.24 \pm 2.00$ | $2.51 \pm 3.64$ | $0.24 \pm 0.23$ | $0.17 \pm 0.21$ |
| | 1000 | $0.23 \pm 0.05$ | $0.13 \pm 0.07$ | $\mathbf{0.13 \pm 0.26}$ | $5.14 \pm 2.14$ | $2.52 \pm 3.55$ | $0.20 \pm 0.23$ | $0.17 \pm 0.21$ |
| Stove | 50 | $5.81 \pm 2.31$ | $4.10 \pm 2.08$ | $10.01 \pm 8.91$ | $3.87 \pm 2.55$ | $1.77 \pm 2.56$ | $2.41 \pm 2.70$ | $\mathbf{1.05 \pm 0.64}$ |
| | 100 | $2.88 \pm 1.19$ | $1.83 \pm 0.97$ | $2.53 \pm 6.88$ | $2.96 \pm 1.98$ | $1.66 \pm 2.55$ | $0.98 \pm 0.75$ | $\mathbf{0.59 \pm 0.53}$ |
| | 300 | $0.89 \pm 0.35$ | $0.40 \pm 0.16$ | $1.00 \pm 5.31$ | $2.70 \pm 2.25$ | $1.52 \pm 2.41$ | $0.45 \pm 0.49$ | $\mathbf{0.37 \pm 0.39}$ |
| | 1000 | $0.26 \pm 0.10$ | $\mathbf{0.18 \pm 0.12}$ | $0.19 \pm 0.27$ | $2.55 \pm 1.83$ | $1.40 \pm 2.36$ | $0.35 \pm 0.37$ | $0.40 \pm 0.49$ |
| Table | 50 | $4.43 \pm 1.63$ | $2.47 \pm 1.49$ | $5.55 \pm 5.29$ | $7.11 \pm 4.14$ | $6.78 \pm 5.05$ | $1.66 \pm 5.01$ | $\mathbf{0.78 \pm 1.05}$ |
| | 100 | $1.99 \pm 0.72$ | $1.04 \pm 0.65$ | $1.60 \pm 3.47$ | $6.76 \pm 3.98$ | $6.47 \pm 5.03$ | $0.66 \pm 1.40$ | $\mathbf{0.38 \pm 0.70}$ |
| | 300 | $0.60 \pm 0.23$ | $\mathbf{0.29 \pm 0.19}$ | $0.61 \pm 2.18$ | $6.41 \pm 3.73$ | $6.25 \pm 4.92$ | $0.41 \pm 1.60$ | $0.32 \pm 1.15$ |
| | 1000 | $0.18 \pm 0.07$ | $\mathbf{0.15 \pm 0.13}$ | $0.40 \pm 1.44$ | $6.21 \pm 3.69$ | $6.19 \pm 4.88$ | $0.42 \pm 3.71$ | $0.31 \pm 1.00$ |
| Telephone | 50 | $3.43 \pm 0.85$ | $1.79 \pm 0.77$ | $19.34 \pm 10.40$ | $5.92 \pm 3.27$ | $0.73 \pm 1.67$ | $0.57 \pm 1.02$ | $\mathbf{0.23 \pm 0.30}$ |
| | 100 | $1.75 \pm 0.41$ | $0.91 \pm 0.38$ | $3.86 \pm 7.16$ | $4.30 \pm 2.75$ | $0.72 \pm 1.70$ | $0.19 \pm 0.29$ | $\mathbf{0.11 \pm 0.18}$ |
| | 300 | $0.57 \pm 0.12$ | $0.19 \pm 0.10$ | $0.40 \pm 3.23$ | $3.03 \pm 2.26$ | $0.62 \pm 1.42$ | $0.12 \pm 0.24$ | $\mathbf{0.09 \pm 0.16}$ |
| | 1000 | $0.17 \pm 0.03$ | $\mathbf{0.06 \pm 0.07}$ | $0.30 \pm 2.59$ | $2.49 \pm 2.06$ | $0.63 \pm 1.46$ | $0.10 \pm 0.21$ | $0.09 \pm 0.17$ |
| Tower | 50 | $3.51 \pm 2.07$ | $3.02 \pm 2.13$ | $11.13 \pm 13.40$ | $5.70 \pm 4.04$ | $2.36 \pm 2.03$ | $1.61 \pm 1.25$ | $\mathbf{0.92 \pm 0.82}$ |
| | 100 | $1.71 \pm 0.97$ | $1.33 \pm 0.85$ | $2.07 \pm 6.96$ | $4.65 \pm 3.32$ | $1.62 \pm 1.66$ | $0.85 \pm 0.76$ | $\mathbf{0.75 \pm 0.78}$ |
| | 300 | $0.50 \pm 0.31$ | $0.40 \pm 0.30$ | $\mathbf{0.19 \pm 0.25}$ | $3.64 \pm 2.87$ | $1.60 \pm 1.51$ | $0.50 \pm 0.52$ | $0.45 \pm 0.46$ |
| | 1000 | $\mathbf{0.15 \pm 0.08}$ | $0.20 \pm 0.18$ | $0.21 \pm 0.29$ | $3.88 \pm 3.57$ | $1.47 \pm 1.64$ | $0.46 \pm 0.47$ | $0.44 \pm 0.53$ |
| Train | 50 | $2.37 \pm 1.09$ | $2.11 \pm 1.12$ | $2.29 \pm 3.56$ | $5.83 \pm 2.15$ | $3.68 \pm 2.58$ | $0.65 \pm 0.54$ | $\mathbf{0.37 \pm 0.30}$ |
| | 100 | $1.17 \pm 0.53$ | $1.00 \pm 0.53$ | $0.21 \pm 0.20$ | $5.83 \pm 1.81$ | $3.38 \pm 2.53$ | $0.39 \pm 0.43$ | $0.22 \pm 0.18$ |
| | 300 | $0.36 \pm 0.17$ | $0.26 \pm 0.14$ | $\mathbf{0.09 \pm 0.08}$ | $5.63 \pm 2.04$ | $3.24 \pm 2.43$ | $0.23 \pm 0.21$ | $0.22 \pm 0.20$ |
| | 1000 | $0.11 \pm 0.05$ | $0.11 \pm 0.09$ | $\mathbf{0.09 \pm 0.08}$ | $6.46 \pm 2.69$ | $2.83 \pm 2.28$ | $0.22 \pm 0.22$ | $0.21 \pm 0.18$ |
| Vessel | 50 | $2.07 \pm 0.93$ | $1.61 \pm 0.93$ | $5.27 \pm 8.20$ | $4.99 \pm 2.18$ | $3.23 \pm 2.34$ | $1.03 \pm 1.88$ | $\mathbf{0.54 \pm 0.51}$ |
| | 100 | $1.00 \pm 0.45$ | $0.85 \pm 0.48$ | $1.38 \pm 4.73$ | $5.05 \pm 2.14$ | $2.98 \pm 2.18$ | $0.68 \pm 0.89$ | $\mathbf{0.35 \pm 0.37}$ |
| | 300 | $0.31 \pm 0.15$ | $\mathbf{0.29 \pm 0.23}$ | $0.62 \pm 2.47$ | $4.90 \pm 2.02$ | $2.73 \pm 2.00$ | $0.57 \pm 1.02$ | $0.33 \pm 0.42$ |
| | 1000 | $\mathbf{0.10 \pm 0.04}$ | $0.18 \pm 0.21$ | $0.46 \pm 1.42$ | $5.21 \pm 2.11$ | $2.55 \pm 1.93$ | $0.50 \pm 0.75$ | $0.33 \pm 0.37$ |
| Washer | 50 | $8.44 \pm 1.32$ | $6.14 \pm 2.10$ | $7.44 \pm 4.24$ | $2.48 \pm 0.57$ | $\mathbf{0.38 \pm 0.36}$ | $2.66 \pm 1.10$ | $1.41 \pm 0.64$ |
| | 100 | $4.02 \pm 0.42$ | $2.66 \pm 0.76$ | $1.35 \pm 2.15$ | $2.15 \pm 0.71$ | $\mathbf{0.36 \pm 0.44}$ | $1.26 \pm 0.87$ | $0.97 \pm 0.66$ |
| | 300 | $1.27 \pm 0.16$ | $0.49 \pm 0.13$ | $\mathbf{0.11 \pm 0.10}$ | $1.72 \pm 0.40$ | $0.32 \pm 0.41$ | $0.25 \pm 0.26$ | $0.49 \pm 0.56$ |
| | 1000 | $0.36 \pm 0.05$ | $0.13 \pm 0.08$ | $\mathbf{0.09 \pm 0.08}$ | $1.65 \pm 0.30$ | $0.25 \pm 0.35$ | $0.18 \pm 0.18$ | $0.46 \pm 0.49$ |

Table 6: Results for each ShapeNet novel category

| Category | Context | Method | | | | | | |
|---|---|---|---|---|---|---|---|---|
| | | Gauss densification | PCN | OccNet | DeepSDF | IGR | MSC (w/o opt) | MSC (w/ opt) |
| Bag | 50 | 4.55 ± 1.29 | 3.47 ± 1.49 | 6.22 ± 7.01 | 5.11 ± 2.49 | 1.63 ± 1.38 | 2.48 ± 2.39 | **1.08 ± 0.63** |
| | 100 | 2.21 ± 0.59 | 1.79 ± 0.62 | 0.85 ± 2.88 | 4.02 ± 2.14 | 1.54 ± 1.39 | 1.37 ± 1.37 | **0.74 ± 0.51** |
| | 300 | 0.67 ± 0.21 | 0.64 ± 0.32 | **0.24 ± 0.21** | 3.52 ± 1.80 | 1.37 ± 1.30 | 0.86 ± 1.46 | 0.57 ± 0.40 |
| | 1000 | **0.19 ± 0.06** | 0.37 ± 0.23 | 0.27 ± 0.22 | 2.95 ± 1.69 | 1.38 ± 1.27 | 0.65 ± 0.49 | 0.57 ± 0.50 |
| Bed | 50 | 5.35 ± 1.52 | 3.79 ± 1.56 | 4.64 ± 6.41 | 4.26 ± 2.15 | 3.72 ± 2.99 | 2.28 ± 1.94 | **1.33 ± 0.94** |
| | 100 | 2.69 ± 0.74 | 1.85 ± 0.69 | 1.51 ± 4.69 | 4.13 ± 2.08 | 3.32 ± 2.89 | 1.51 ± 1.40 | **0.99 ± 0.86** |
| | 300 | 0.88 ± 0.26 | 0.74 ± 0.35 | **0.64 ± 1.06** | 4.06 ± 2.12 | 3.17 ± 3.00 | 1.07 ± 0.86 | 0.91 ± 0.78 |
| | 1000 | **0.26 ± 0.08** | 0.51 ± 0.34 | 0.65 ± 0.84 | 3.97 ± 2.22 | 3.06 ± 2.93 | 1.02 ± 0.89 | 0.92 ± 0.84 |
| Bench | 50 | 2.97 ± 1.18 | 1.78 ± 1.02 | 7.23 ± 7.15 | 9.12 ± 3.86 | 4.18 ± 2.72 | 1.32 ± 2.03 | **0.63 ± 0.52** |
| | 100 | 1.38 ± 0.57 | 0.78 ± 0.46 | 1.97 ± 4.33 | 9.19 ± 3.93 | 3.87 ± 2.63 | 0.71 ± 1.50 | **0.42 ± 0.49** |
| | 300 | 0.40 ± 0.18 | **0.27 ± 0.16** | 0.39 ± 1.36 | 9.14 ± 4.00 | 3.76 ± 2.59 | 0.45 ± 0.48 | 0.36 ± 0.41 |
| | 1000 | **0.12 ± 0.05** | 0.17 ± 0.13 | 0.26 ± 0.65 | 9.09 ± 4.10 | 3.78 ± 2.54 | 0.42 ± 0.86 | 0.35 ± 0.40 |
| Birdhouse | 50 | 5.07 ± 1.82 | 3.69 ± 1.68 | 3.83 ± 5.30 | 3.99 ± 3.32 | 2.30 ± 1.75 | 2.13 ± 1.17 | **1.27 ± 0.68** |
| | 100 | 2.52 ± 0.90 | 1.90 ± 0.82 | 0.93 ± 2.47 | 3.76 ± 3.49 | 2.05 ± 1.55 | 1.24 ± 0.74 | **0.69 ± 0.48** |
| | 300 | 0.79 ± 0.28 | 0.53 ± 0.26 | **0.23 ± 0.25** | 3.35 ± 3.00 | 1.85 ± 1.43 | 0.68 ± 0.46 | 0.57 ± 0.46 |
| | 1000 | **0.24 ± 0.08** | 0.29 ± 0.18 | 0.26 ± 0.21 | 3.25 ± 3.29 | 1.92 ± 1.49 | 0.60 ± 0.41 | 0.53 ± 0.39 |
| Bookshelf | 50 | 4.94 ± 1.64 | 2.91 ± 1.44 | 11.14 ± 8.91 | 4.36 ± 2.67 | 1.98 ± 2.89 | 1.92 ± 1.37 | **1.06 ± 0.56** |
| | 100 | 2.42 ± 0.76 | 1.50 ± 0.67 | 2.41 ± 4.42 | 3.58 ± 2.49 | 1.82 ± 2.65 | 1.16 ± 0.83 | **0.79 ± 0.44** |
| | 300 | 0.78 ± 0.26 | 0.60 ± 0.32 | **0.53 ± 1.25** | 2.86 ± 2.44 | 1.78 ± 2.71 | 0.88 ± 0.58 | 0.69 ± 0.46 |
| | 1000 | **0.24 ± 0.08** | 0.42 ± 0.30 | 0.47 ± 0.53 | 2.65 ± 2.28 | 1.76 ± 2.70 | 0.81 ± 0.52 | 0.66 ± 0.44 |
| Bus | 50 | 3.69 ± 1.04 | 3.30 ± 1.12 | 3.81 ± 4.88 | 6.06 ± 1.76 | 2.39 ± 1.62 | 0.86 ± 0.69 | **0.53 ± 0.40** |
| | 100 | 1.79 ± 0.50 | 1.58 ± 0.55 | 0.41 ± 1.15 | 6.01 ± 1.64 | 1.93 ± 1.50 | 0.35 ± 0.32 | **0.30 ± 0.28** |
| | 300 | 0.55 ± 0.15 | 0.31 ± 0.13 | **0.10 ± 0.18** | 5.86 ± 1.47 | 1.54 ± 1.28 | 0.18 ± 0.22 | 0.24 ± 0.23 |
| | 1000 | 0.16 ± 0.04 | **0.09 ± 0.08** | 0.09 ± 0.13 | 5.66 ± 1.40 | 1.35 ± 1.21 | 0.15 ± 0.19 | 0.25 ± 0.22 |
| Cap | 50 | 3.61 ± 0.78 | 3.91 ± 1.81 | 6.01 ± 4.08 | 3.95 ± 1.24 | 3.45 ± 1.89 | 2.31 ± 1.06 | **1.24 ± 0.62** |
| | 100 | 1.67 ± 0.34 | 1.58 ± 0.64 | 4.09 ± 3.45 | 3.77 ± 1.16 | 3.08 ± 1.74 | 1.71 ± 0.94 | **0.82 ± 0.50** |
| | 300 | 0.49 ± 0.09 | **0.47 ± 0.13** | 3.32 ± 3.16 | 3.69 ± 0.98 | 2.94 ± 1.76 | 1.58 ± 0.77 | 0.67 ± 0.54 |
| | 1000 | **0.14 ± 0.03** | 0.28 ± 0.10 | 2.39 ± 2.35 | 3.65 ± 1.07 | 2.90 ± 1.91 | 1.52 ± 0.77 | 0.52 ± 0.38 |
| Keyboard | 50 | 2.23 ± 0.60 | 1.03 ± 0.72 | 21.36 ± 14.61 | 6.04 ± 2.61 | 1.27 ± 0.87 | 2.53 ± 16.75 | **0.29 ± 0.51** |
| | 100 | 1.11 ± 0.30 | 0.47 ± 0.31 | 6.00 ± 10.78 | 6.84 ± 2.82 | 1.08 ± 0.90 | 0.56 ± 1.64 | **0.17 ± 0.25** |
| | 300 | 0.38 ± 0.10 | 0.24 ± 0.24 | 1.47 ± 6.24 | 7.55 ± 2.84 | 1.02 ± 0.79 | 0.43 ± 1.27 | **0.19 ± 0.36** |
| | 1000 | **0.12 ± 0.03** | 0.14 ± 0.26 | 0.40 ± 2.31 | 9.05 ± 3.01 | 1.02 ± 0.85 | 0.56 ± 2.86 | 0.18 ± 0.31 |
| Earphone | 50 | 2.75 ± 1.05 | 2.50 ± 1.15 | 9.06 ± 13.27 | 6.08 ± 3.39 | 7.05 ± 4.08 | 8.05 ± 9.19 | **2.20 ± 2.40** |
| | 100 | **1.30 ± 0.53** | 1.79 ± 0.83 | 4.95 ± 12.48 | 5.84 ± 3.70 | 6.73 ± 4.19 | 4.47 ± 4.42 | 1.80 ± 1.93 |
| | 300 | **0.40 ± 0.19** | 1.53 ± 0.85 | 2.71 ± 9.27 | 5.28 ± 3.36 | 6.61 ± 4.04 | 5.54 ± 11.01 | 1.49 ± 1.50 |
| | 1000 | **0.12 ± 0.06** | 1.42 ± 0.83 | 2.12 ± 7.08 | 5.30 ± 3.47 | 6.54 ± 4.50 | 3.99 ± 7.08 | 1.34 ± 1.45 |
| Guitar | 50 | 0.95 ± 0.42 | 0.48 ± 0.32 | 47.72 ± 19.57 | 11.76 ± 3.77 | 5.88 ± 2.25 | 2.22 ± 11.36 | **0.38 ± 0.43** |
| | 100 | 0.42 ± 0.18 | **0.25 ± 0.13** | 25.87 ± 22.62 | 10.92 ± 3.90 | 5.78 ± 2.16 | 1.05 ± 5.93 | 0.30 ± 0.36 |
| | 300 | **0.13 ± 0.05** | 0.13 ± 0.05 | 6.90 ± 13.83 | 9.96 ± 3.92 | 5.70 ± 2.01 | 0.55 ± 0.99 | 0.27 ± 0.30 |
| | 1000 | **0.05 ± 0.01** | 0.11 ± 0.05 | 2.69 ± 9.23 | 9.51 ± 3.91 | 5.67 ± 2.10 | 0.43 ± 0.54 | 0.29 ± 0.31 |
| Microphone | 50 | 1.90 ± 2.19 | 1.82 ± 1.20 | 13.33 ± 22.51 | 11.10 ± 7.32 | 10.58 ± 15.13 | 3.91 ± 6.55 | **1.36 ± 2.36** |
| | 100 | **0.76 ± 0.55** | 1.14 ± 0.90 | 6.16 ± 16.34 | 10.90 ± 8.04 | 9.70 ± 14.00 | 3.99 ± 7.72 | 0.94 ± 1.46 |
| | 300 | **0.22 ± 0.16** | 0.75 ± 0.98 | 2.13 ± 6.77 | 9.81 ± 7.40 | 9.92 ± 14.17 | 2.90 ± 7.11 | 0.90 ± 1.21 |
| | 1000 | **0.07 ± 0.04** | 0.64 ± 0.94 | 1.51 ± 5.28 | 9.64 ± 7.53 | 9.84 ± 14.61 | 1.72 ± 2.55 | 0.92 ± 1.67 |
| Motorcycle | 50 | 2.42 ± 0.61 | 1.90 ± 0.71 | 1.16 ± 1.33 | 5.09 ± 1.18 | 2.40 ± 1.36 | 1.72 ± 0.94 | **1.04 ± 0.42** |
| | 100 | 1.21 ± 0.30 | 1.03 ± 0.38 | **0.39 ± 0.16** | 5.02 ± 1.15 | 2.08 ± 1.23 | 1.22 ± 0.60 | 0.81 ± 0.38 |
| | 300 | 0.40 ± 0.10 | 0.52 ± 0.19 | **0.33 ± 0.14** | 4.98 ± 1.08 | 1.92 ± 1.12 | 1.03 ± 0.49 | 0.72 ± 0.38 |
| | 1000 | **0.13 ± 0.03** | 0.44 ± 0.17 | 0.39 ± 0.16 | 4.79 ± 1.02 | 1.91 ± 1.11 | 0.96 ± 0.47 | 0.71 ± 0.36 |
| Pistol | 50 | 1.95 ± 0.78 | 1.27 ± 0.61 | 0.43 ± 0.40 | 10.19 ± 2.98 | 7.43 ± 4.01 | 0.59 ± 0.44 | **0.32 ± 0.24** |
| | 100 | 0.95 ± 0.38 | 0.57 ± 0.25 | **0.14 ± 0.17** | 10.10 ± 2.84 | 6.55 ± 3.36 | 0.38 ± 0.28 | 0.19 ± 0.20 |
| | 300 | 0.29 ± 0.11 | 0.19 ± 0.08 | **0.11 ± 0.06** | 9.77 ± 2.59 | 6.11 ± 3.53 | 0.28 ± 0.25 | 0.15 ± 0.29 |
| | 1000 | **0.09 ± 0.03** | 0.12 ± 0.07 | 0.13 ± 0.07 | 9.76 ± 2.67 | 5.78 ± 3.47 | 0.25 ± 0.28 | 0.14 ± 0.14 |
| Remote Control | 50 | 2.90 ± 1.08 | 1.81 ± 0.94 | 16.70 ± 14.58 | 6.55 ± 3.74 | 1.68 ± 1.68 | 0.58 ± 0.65 | **0.23 ± 0.30** |
| | 100 | 1.48 ± 0.50 | 0.85 ± 0.40 | 0.98 ± 3.40 | 5.30 ± 3.44 | 1.52 ± 1.57 | 0.21 ± 0.26 | **0.13 ± 0.17** |
| | 300 | 0.46 ± 0.16 | 0.19 ± 0.11 | **0.06 ± 0.02** | 4.13 ± 2.93 | 1.57 ± 1.58 | 0.10 ± 0.09 | 0.14 ± 0.12 |
| | 1000 | 0.14 ± 0.04 | 0.06 ± 0.05 | 0.07 ± 0.02 | 3.91 ± 3.06 | 1.59 ± 1.66 | 0.09 ± 0.09 | **0.06 ± 0.05** |
| Skateboard | 50 | 1.51 ± 0.46 | 0.89 ± 0.41 | 8.69 ± 12.50 | 6.34 ± 3.37 | 4.76 ± 2.59 | 0.94 ± 0.97 | **0.39 ± 0.24** |
| | 100 | 0.70 ± 0.23 | 0.44 ± 0.24 | 1.36 ± 6.12 | 6.03 ± 3.12 | 4.27 ± 2.57 | 0.98 ± 5.22 | **0.29 ± 0.28** |
| | 300 | 0.22 ± 0.08 | **0.18 ± 0.10** | 0.42 ± 3.14 | 6.40 ± 3.12 | 3.87 ± 2.21 | 0.49 ± 0.41 | 0.23 ± 0.35 |
| | 1000 | **0.08 ± 0.02** | 0.14 ± 0.09 | 0.31 ± 1.71 | 7.13 ± 3.69 | 3.64 ± 1.95 | 0.45 ± 0.33 | 0.21 ± 0.19 |

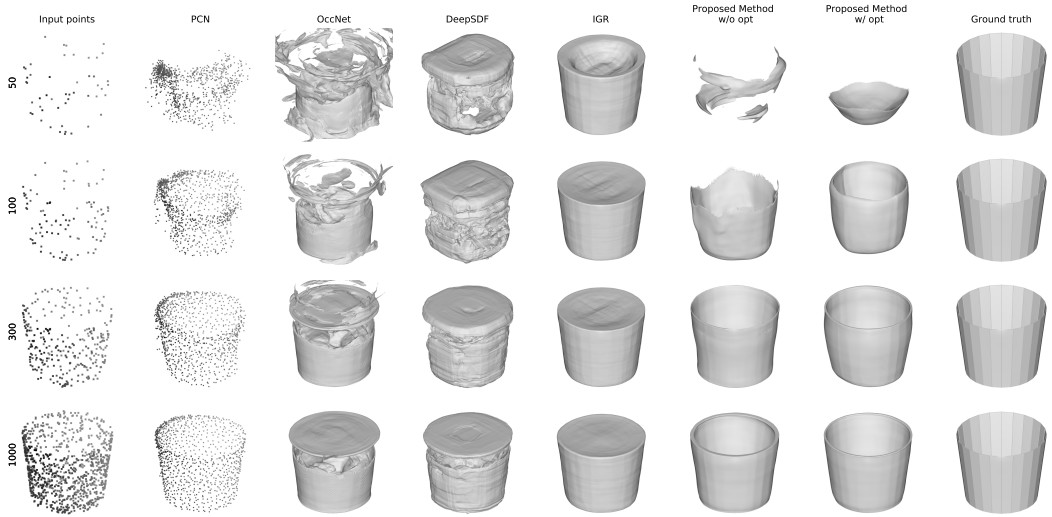

Figure 7: **Shape completion results on ShapeNet for *Basket*** (training category). Our method starts conservatively and improves as more points are added (it also correctly infers that there is no lid).

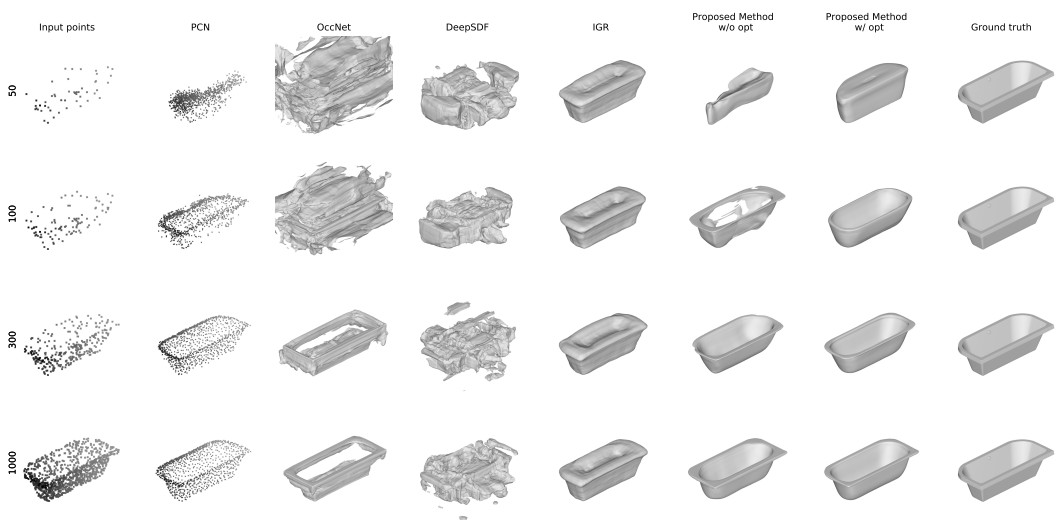

Figure 8: **Shape completion results on ShapeNet for *Bathtub*** (training category). Our proposed method captures the large cavity with only 100 observations, while both DeepSDF and OccNet produce artifacts at this level of sparsity.

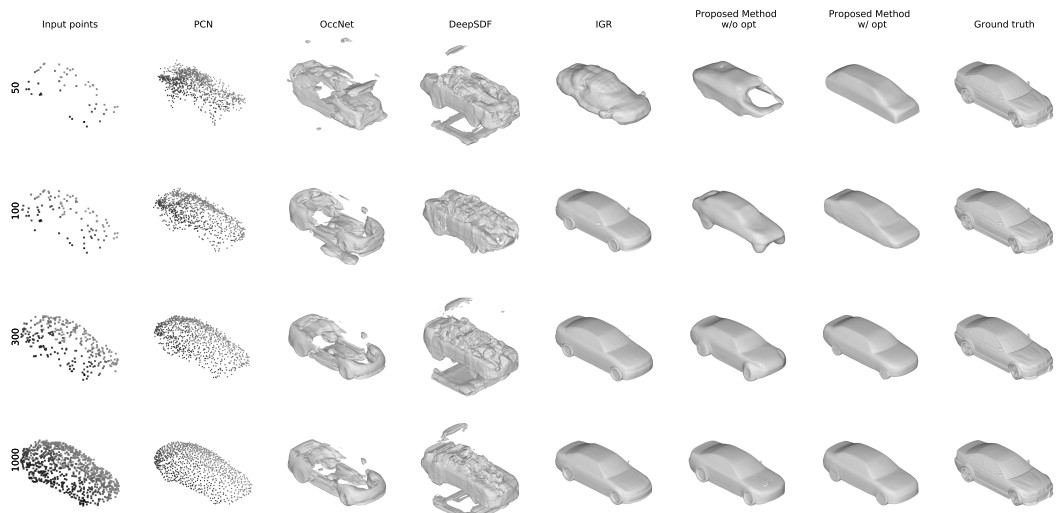

Figure 9: **Shape completion results on ShapeNet for *Car*** (training category). Our method produces a smooth car-like surface with only 50 observations, while both DeepSDF and OccNet generate large artifacts at this level of sparsity. Additionally, PCN fails to produce a car-like point cloud from only 50 observations.

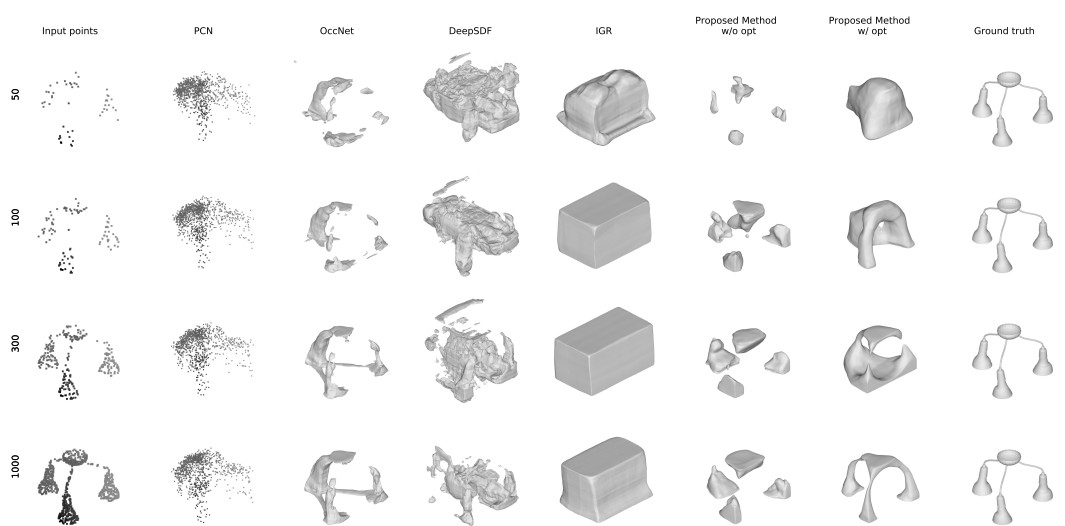

Figure 10: **Shape completion results on ShapeNet for *Lamp*** (training category). All methods struggle with predictions for this category, irrespective of the number of observations. Particularly, IGR consistently assumes that the target object is rectangular, even at higher density levels. At these levels, our method succeeds in capturing the approximate geometry of the lamp.

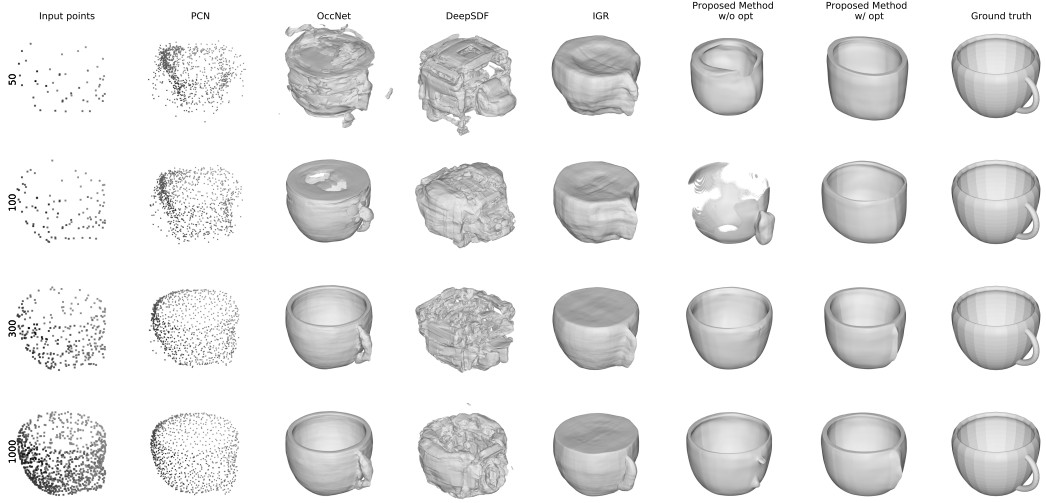

Figure 11: **Shape completion results on ShapeNet for *Mug*** (training category). Our method is the only one that succeeds in capturing the cavity of the mug from sparse observations, while other methods assume there is a lid. As expected, PCN only produces what looks like a densified version of the original point cloud.

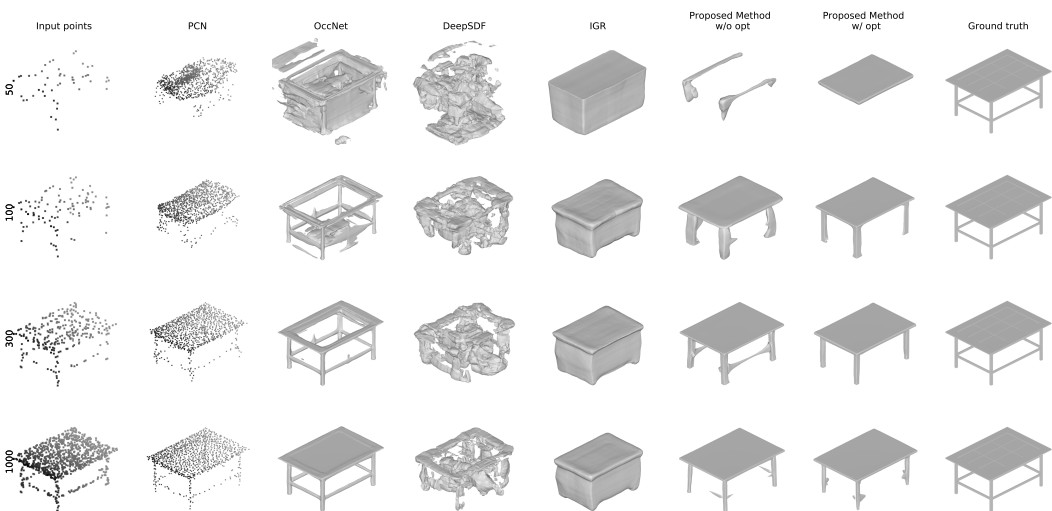

Figure 12: **Shape completion results on ShapeNet for *Table*** (training category). Our method correctly captures the rectangular portion of the table when only 50 observations are available, and gradually completes other parts as the number of observations increase. IGR, on the other hand, assumes that the object is box-shaped at lower densities and maintains this assumption even at higher density levels.

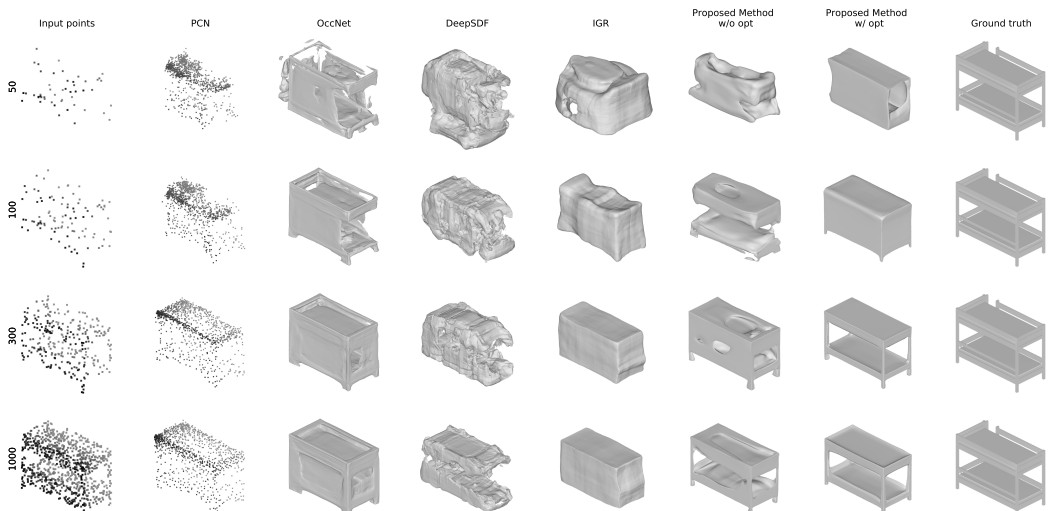

Figure 13: **Shape completion results on ShapeNet for** *Bed* (novel category). Our proposed method correctly predicts the overall box shape of the object, and gradually improves this prediction as more observations points become available. Other methods, such as OccNet and IGR, fail to separate the top and lower portions of the object even at higher density levels.

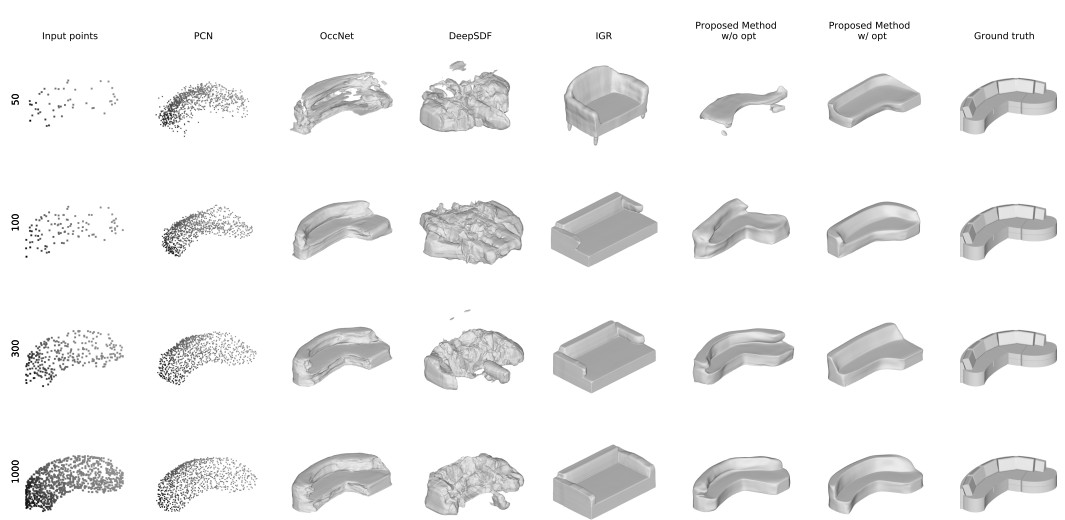

Figure 14: **Shape completion results on ShapeNet for** *Bed* (novel category). With only 50 points our proposed method is already capable of predicting the correct shape, that is then further refined with more observations. OccNet requires a higher number of observations before it settles on the correct shape, while IGR converges to the wrong object.

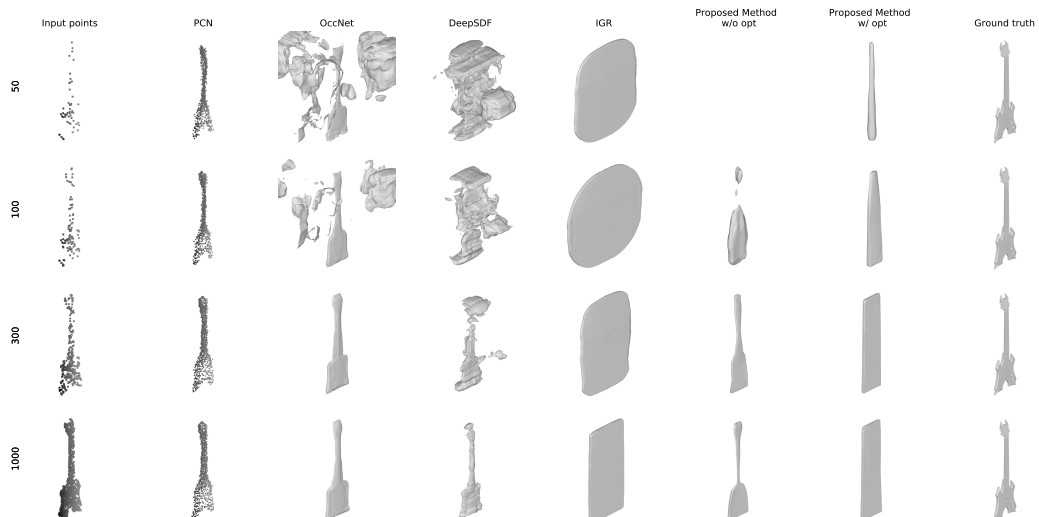

Figure 15: **Shape completion results on ShapeNet for *Guitar*** (novel category). Here, PCN is the only method that seems to succeed at shape completion, mostly because of the compact nature of the object that enables accurate densification. Interestingly, as the number of observations increase, our proposed method performs better without post-encoder optimization, producing a reasonable output with 300 observations.

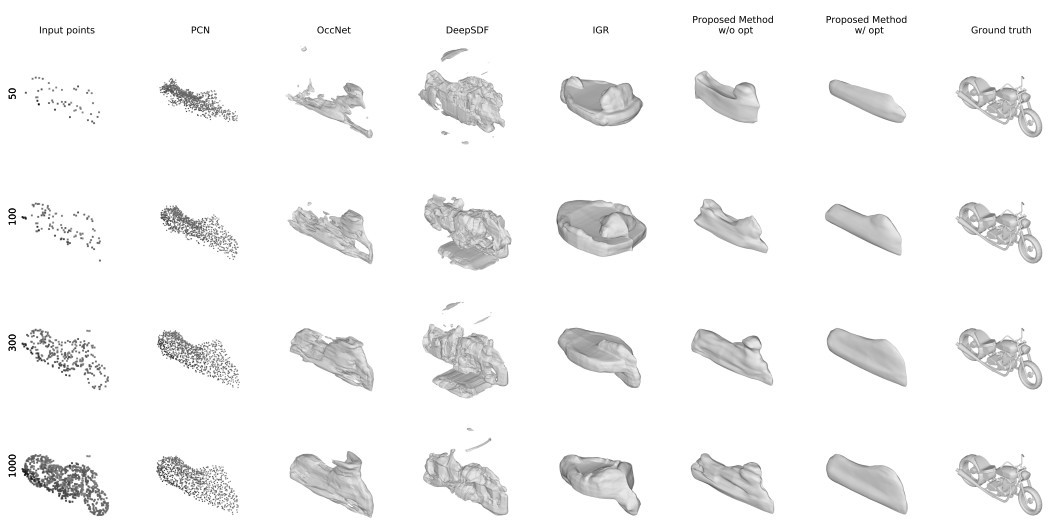

Figure 16: **Shape completion results on ShapeNet for *Motorcycle*** (novel category). Similar to the *guitar* category (Figure 10), our proposed method performs better without post-encoder optimization. We attribute this to a smoothing effect generated by this step, that enables a better generalization at the expense of finer details.

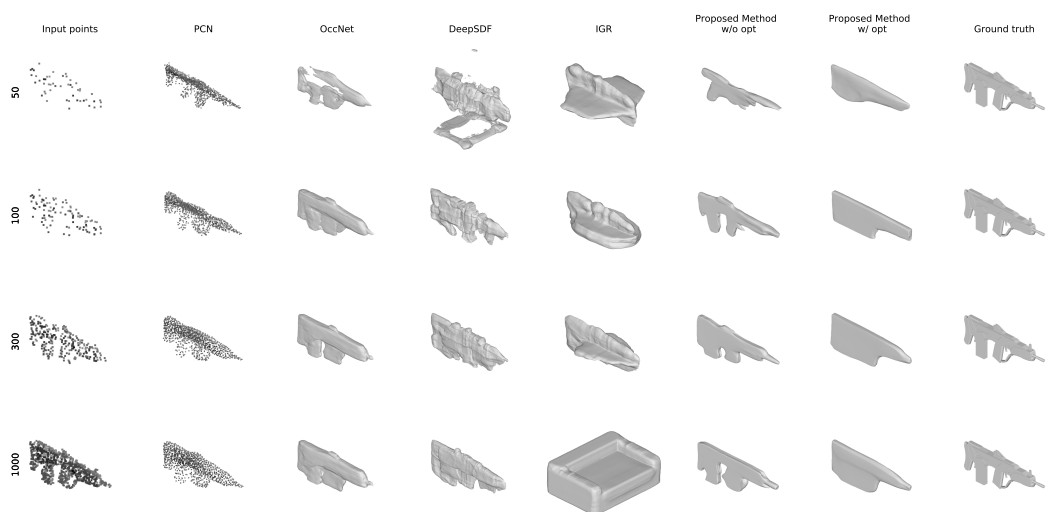

Figure 17: **Shape completion results on ShapeNet for *Pistol*** (novel category). Interestingly, IGR seems to change the assumed category as the number of observation points increase. Again, due to the presence of finer details our method performs better without post-encoder optimization, achieving reasonable results with as few as 100 observations.

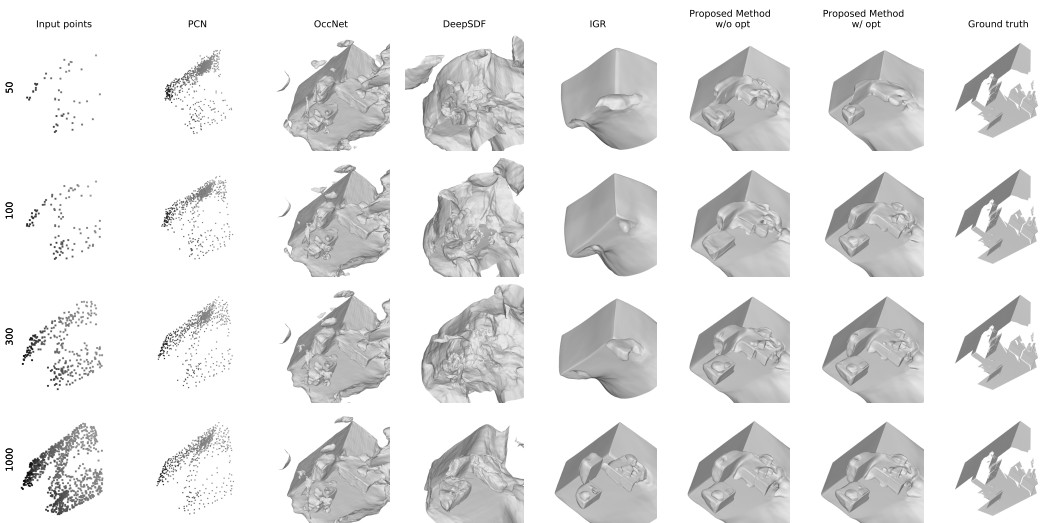

Figure 18: **ICL-NUIM shape completion results for *office rooms***. With as few as 50 points, our proposed method is able to recover most details of the room, including furniture arrangement, wall distribution and floor. As more observations are introduced, these details are further refined without diverging. PCN results are mostly a densification of the observed points, IGR requires higher densities for a proper convergence, and DeepSDF produces a large amount of artifacts throughout the scene.

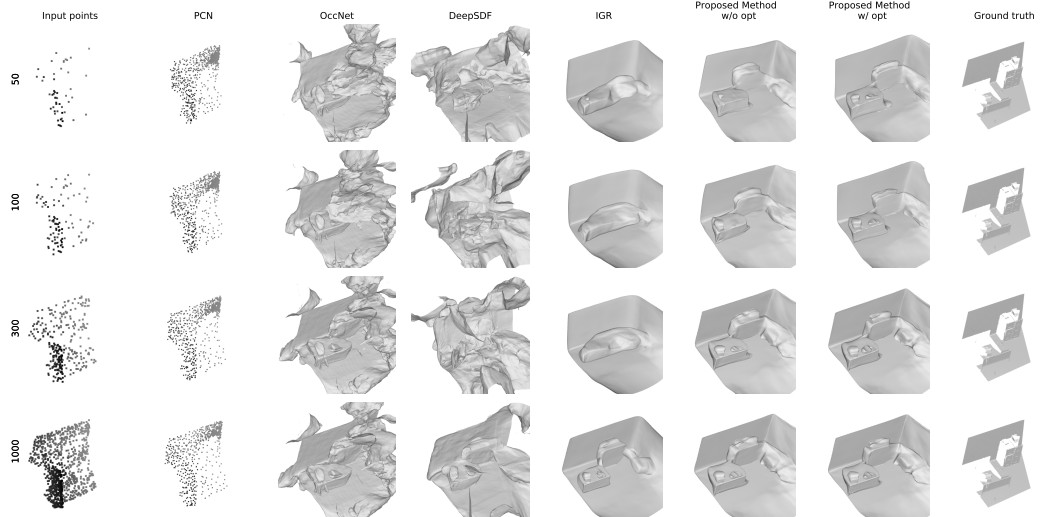

Figure 19: **ICL-NUIM shape completion results for *office rooms***. With as few as 50 points, our proposed method is able to recover most details of the room, including furniture arrangement, wall distribution and floor. As more observations are introduced, these details are further refined without diverging. PCN results are mostly a densification of the observed points, IGR requires higher densities for a proper convergence, and DeepSDF produces a large amount of artifacts throughout the scene.

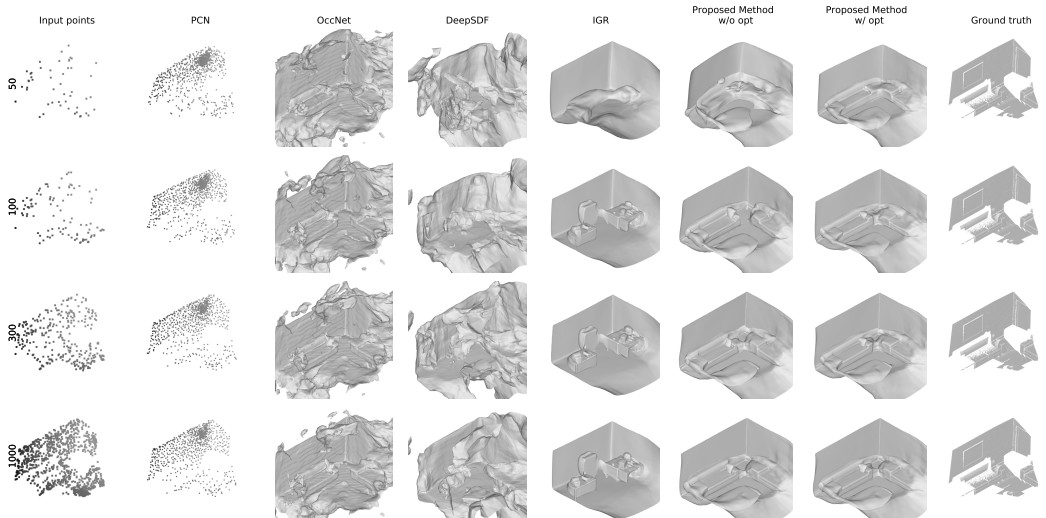

Figure 20: **ICL-NUIM shape completion results for *living rooms***. With as few as 50 points, our proposed method is able to recover most details of the room, including furniture arrangement, wall distribution and floor. As more observations are introduced, these details are further refined without diverging. PCN results are mostly a densification of the observed points, IGR requires higher densities for a proper convergence, and DeepSDF produces a large amount of artifacts throughout the scene.

