# OpenReview forum: "Bayesian Meta-Learning for Few-Shot 3D Shape Completion "
_ICLR.cc/2021/Conference — Reject_

### Official Review · AnonReviewer2 · 2020-10-21
**Interesting idea and impressive results, but lack of literature review.**

**Rating:** 7
**Confidence:** 4

**Review:**

This paper introduces a meta-learning approach for the neural implicit representation of 3D shapes. The main idea, in my understanding, is to consider the points in the input point cloud as few-shot examples of the object so that each of them can be encoded in a way to best approximate the entire object information. The experiments show that the network can reconstruct the entire shape well even with a very small number of the input points, such as 50 and 100. For better reconstruction, the authors also proposed to use some implicit function regularizations, which are introduced in a previous work (Gropp et al., 2020).


*** Strengths ***
To my knowledge, this idea of using the meta-learning technique in the neural implicit representation is novel, except for the following concurrent work (I hope the authors discuss this work in the revision):
Sitzmann et al., MetaSDF: Meta-learning Signed Distance Functions, arXiv:2006.09662.

Also, the experiments show impressive results both quantitatively and qualitatively. It’s interesting to see that the proposed method generates a plausible output even with 50 input points.


*** Weaknesses ***
All the formulations directly come from previous work; Maeda et al., 2020 for the meta-learning formulation, and Gropp et al., 2020 for the implicit function regularizers. Although, I think it’s still worth showing how the meta-learning technique can be applied in this implicit function learning problem.

There are many recent works about 3D shape completion and point cloud upsampling, but the authors only referred to and compared with PCN (Yuan et al., 2018): To name a few:

[Completion]
Huang et al., PF-Net: Point Fractal Network for 3D Point Cloud Completion, CVPR 2020.
Liu et al., Morphing and Sampling Network for Dense Point Cloud Completion, AAAI 2020

[Upsampling]
Yu et al., PU-Net: Point Cloud Upsampling Network, CVPR 2018
(This work has 120 citations, meaning that there are many other recent works about deep upsampling).
Li et al., PU-GAN: a Point Cloud Upsampling Adversarial Network, ICCV 2019.


*** Justification ***
I like this idea of using meta-learning in shape completion/upsampling. But the lack of literature review on this problem and comparisons looks like a big weak point. I hope the authors address this concern in the revision.


*** Discussion / Comments ***
I guess the proposed idea can be applied to the other types of input data. For instance, one can consider a multi-view or sketch-based 3D reconstruction network which, however, can still reconstruct the shape well even with a single image or a few strokes.

In Figure 1, ShapeNet40 -> ShapeNet?
It would be clearer to show one encoder with variable ‘m’ for the input dimension of the last MLP layer and mention in the caption that the ‘m’ is 512 for ShapeNet and 256 for ICL-NUIM.

---

> ### Author Response · Authors · 2020-11-20
> **Response to Reviewer #2**
>
> Thank you for pointing out the additional related work, we will include these in the revised version and provide additional explanations. One of the major differences between the works mentioned and ours is that they complete the shape from partial observations that are locally dense. They also use an hierarchical encoder and decoder in order to separately extract global and local information. Meanwhile, our method aims to leverage the ability of meta-learning to complete the shape from uniformly distributed, super-sparse observations, from which it can be difficult to extract local information. The related work on upsampling deals with a setting that is more similar to our work.  However, they do not upsample from as sparse a point cloud as the ones used in our study (10~100). PU-Net, for example, upsamples from 5000 points,  and PU-GAN generates a dense point cloud from 256 points.

---

### Official Review · AnonReviewer4 · 2020-10-31
**Use of meta-learning rather confusing here, and relationship to VAE-based encoders not clear**

**Rating:** 4
**Confidence:** 4

**Review:**

This paper proposes a way of reconstructing a surface from sparse point clouds via a "meta learning" approach. Specifically, the authors view each shape in a collection as a "domain", and predicting the SDF values of points in R^3 to reconstruct a given shape (the reconstructed surface is the isosurface of the SDF field) as the "task" for that domain. Then, they use a network to predict a distribution over "task-specific" latent vectors that characterizes the reconstruction task for a given shape. Given a latent-vector sampled for one shape, they pass it to a decoder that predicts the SDF value at any point in R^3 for that shape.

The basic decoder setup follows that of IGR. Specifically, by adding Eikonal (and normal vector) regularization, the authors can train purely on point clouds, without ground truth surfaces or SDF values. (This feature, which is clearly stated in the IGR paper, is quite obfuscated in this paper and should be stated much more clearly, though it is not a contribution of the current work per se.)

I am both intrigued and confused by the meta-learning aspect of this work. While I agree the formal mathematical structure of meta-learning applies to the interpretation that each shape is a task/domain, it seems quite non-standard. I would have expected, say, a collection of shapes from one category to be a "domain", or the tasks to be much less related e.g. one task could be reconstruction and another could be classification. Further, given the "each-shape-is-a-domain" interpretation, the final network architecture ends up looking suspiciously like a VAE, where the encoder (say with a PointNet or similar architecture) takes a point cloud of arbitrary size as input and outputs a local gaussian distribution over latent vectors, and an implicit field decoder that acts exactly as in this paper, taking as input a latent vector sampled from the encoder output. Any difference from a standard-issue VAE, then, boils down to the exact way the encoder's gaussian is computed (and possibly also the training loss, but here the usual ELBO loss appears to be used). In this paper, Eq (2) describes the gaussian computation, drawing from prior work [Maeda et al. 2020].

I am certainly not an expert on meta-learning of any sort, so I cannot comment on the advantages of the proposed approach over a standard VAE encoder. However, the similarities do seem to warrant an explicit comparison, where the overall architecture is as similar as possible (the current encoder with parallel MLPs with shared weights followed by an aggregator appears very similar    to PointNet, so something on those lines could be used), but the encoder explicitly outputs the \mu and \sigma of a gaussian. Indeed, OccNet mentions a generative variant along these lines. I suggest posing this as an ablation study of the proposed method, though, to eliminate any bias due to other architectural differences.

The following paper published at the same time as OccNet is also closely related (generative implicit field model) and should be cited and possibly compared to:

Chen and Zhang, "Learning implicit fields for generative shape modeling", CVPR 2019

The proposed method also allows sampling several completion candidates, since the encoder outputs a distribution. But such sampling does not seem to be demonstrated anywhere in the paper. What would happen if one only passed the distribution mean to the decoder? What do different candidates look like?

To summarize, the proposed method involves several design decisions that distinguish it from prior work. Some of these are novel though somewhat obfuscated, e.g. the "meta learning" solution; some have been used elsewhere, e.g. learning from point clouds only; and some are not relevant to the stated contributions, e.g. architectural differences other than the bare minimum to get the meta learning formulation to work. I request the authors to more clearly state and justify their contributions, especially via the ablation suggested above.

=======================================================

UPDATE: I thank the authors for their responses to our questions. However, I still cannot recommend acceptance, because of the issue both R3 and I mentioned: the comparison to a more traditional VAE-style encoder, which replaces the "aggregator" of this paper with max-pooling plus some number of FC layers to output a mean and variance (as I mentioned earlier, this is almost exactly PointNet with the last layer outputting mean+variance instead of a single feature vector). I definitely do not buy the assertion that VAEs cannot support variable sized point contexts. Just about any shape encoder can be converted to a VAE encoder by modifying the last layer (and training with an appropriate decoder and loss) -- this includes encoders that handle point clouds of varying size like PointNet. In the exchange with the authors, it was not clear that this point was fully appreciated. Hence, while I appreciate the many interesting aspects of this paper, and the additional results provided by the authors during the discussion session, I remain negative and am slightly lowering my rating to make this clear (if this was a journal submission I would mark it as "major revisions" and ask for additional experiments vs VAE baselines which share the bulk of the proposed architecture other than the different aggregator).

---

> ### Author Response · Authors · 2020-11-20
> **Response to Reviewer #4**
>
> [Advantages over VAE]
>
> It is no coincidence that VAE looks similar to our method. Our encoder and VAE’s encoder differ in what they “encode”.  VAE is a method that learns a probabilistic encoder that maps each observation in the dataset to a latent variable.  Meanwhile, our encoder encodes a size-varying set of observations collected from a given point cloud into the object-specific latent variable.  This is a mechanism that allows us to encode a feature that is specific to point clouds of varying size, and this cannot be done with the encoder of VAE.
>
> [Passing the distribution mean to decoder]
>
> This is indeed a natural question and thank you very much for bringing this up.  Our encoder based on Maeda et al’s work uses the inferred variance to weigh each point in the point cloud differently, and it can produce a latent vector from the point cloud of varying size.   We are currently running an ablation study to verify how the usage of variance information affects the outcome.

---

> > ### Comment · AnonReviewer4 · 2020-11-24
> > **Encoder of VAE**
> >
> > I'm a bit confused. Why can a VAE encoder not encode a point cloud of arbitrary size? If I took, say, PointNet, and made the last layer output a mean and variance, that would serve the purpose, no? VAEs are a general framework, a particular VAE can compute (distributions over) object-level latent vectors perfectly well as has been done in much prior work on shape synthesis.
> >
> > Having a _decoder_ output a variable sized point cloud is truly tricky, but that's not what you're arguing for.
> >
> > PS: Apologies for this rather late response.

---

> > > ### Author Response · Authors · 2020-11-25
> > > **Re: Encoder of VAE**
> > >
> > > The ordinary encoder used in VAE does not have an aggregation mechanism and hence cannot handle variable size contexts. According to Maeda et al., the encoder should be the posterior that uses all available observations and the posterior should represent the uncertainty about the latent variable. It approaches the point estimated by the maximum likelihood estimation when the number of available observations increases to infinity, i.e., the variance of the posterior approaches zero asymptotically. This flexible representation of the uncertainty cannot be attained by the mere average or max pooling aggregation.

---

> > ### Author Response · Authors · 2020-11-25
> > **Passing the distribution mean to the decoder**
> >
> > We also performed an additional ablation study comparing the use of an stochastic sample vs. using the deterministic mean. We can see that feeding the stochastic sample to the decoder during training works better than feeding the deterministic mean:
> >
> > chamfer_distance (×100)
> >
> > | num_context | Proposed Method (stochastic sample)  | Proposed Method (deterministic mean) |
> > | ----------- | ---------------  | ------------------------------------ |
> > | 50          | 0.206(±0.218)             | 0.284(±0.738)                        |
> > | 100         |0.174(±0.186)             | 0.279(±0.824)                        |
> > | 300         | 0.163(±0.180)            | 0.285(±0.903)                        |
> > | 1000        |0.158(±0.177)            | 0.292(±0.930)                        |

---

### Official Review · AnonReviewer3 · 2020-11-02
**Interesting Approach, but concerns on baseline comparisons**

**Rating:** 5
**Confidence:** 4

**Review:**

This paper tackles the task of point cloud completion, aiming to infer an implicit occupancy representation given a sparse input point cloud. Following recent practices, the approach represents the shape via a latent-variable conditioned occupancy function $f_{\phi}(x, h)$ that infers occupancy of a point $x$ given latent $h$. The central task addressed here is to be able to infer a posterior distribution over the latent variable given some observed points $D$ i.e. $p_{\theta}(h|D)$.

Following Maeda etal., this paper uses a specific form of this distribution and trains $(\phi,\theta)$ by optimizing the loss on held out surface points and using additional regularization. In particular, each data-point $x_n$ infers a gaussian distribution over $h$ parametrized by $f_n, g_n$ and these are aggregated to get the full posterior. This compositional nature of prediction (each data point infers a distribution over $h$) in particular helps the approach robustly scale to use more/less points during inference.

**Strengths**
- A requirement for the posterior distribution is that given more point samples, the shape should ideally become more refined and closer to the true shape. The empirical and qualitative results clearly show that this is a property of the proposed approach.

- The empirical results reported are encouraging, and in particular the proposed approach seems to yield more accurate reconstructions compared to vanilla encoders used in prior approaches e.g. OccNet/PCN (although I have some concerns regarding this).

- The paper presents results on Object and Scene reconstruction datasets and shows the ability to recover shapes from just a few (10) to many (1000) samples.


**Concerns/Comments**

- While this approach uses a specific factored form of $p_{\theta}(h|D)$, the more 'typical' way is learn a non-factored encoder (e.g. a PointNet) that directly predicts a variational distribution $q_{\theta}(h|D)$. Two of the baselines reported (PCN, OccNet) do use this approach of using an encoder, but I am concerned the comparison is unfair.
In particular, these baselines are trained using a fixed number of input points in set $D$ (around 5000), but tested using a different number of input points (from 10 to 1000) that are far fewer than the size of $D$ seen in training. Such a discrepancy between training and testing settings can obviously adversely affect the results and might explain why methods like OccNet perform so badly. This is not as much of a concern for the proposed method because of the factored and per-datapoint prediction of $h$. I would therefore really like to see results with the encoders of these baselines trained using a variable number of input points, similar to what would be used at testing. In the current form, the results merely show that encoders trained using many points work poorly when using fewer points for inference.


- The proposed approach parametrizes $p(h|x_n)$ as a gaussian for tractability. I would assume this is a limiting choice (e.g. given a single point, say on a chair back, the possible shape space is arguably multi-modal). I would appreciate some discussion on why the method still works despite this choice.

- Despite being novel in context of shape completion, I am slightly concerned regarding the technical novelty as the overall approach is a direct application of the model proposed in Maeda et. al. That said, the insight that implicit occupancy prediction from sparse point clouds can be viewed as a meta-learning task is still interesting and I might still be in favor of acceptance if all the empirical concerns were addressed.


- (minor concern) I don't think the normalized metrics add much to the evaluation. I would instead encourage the authors to report alternate metrics such as F-score.

----
Overall, while I think the approach is interesting and novel in context of shape completion, I cannot recommend acceptance until the concerns regarding the fair comparisons to baseline methods are addressed.

---

> ### Author Response · Authors · 2020-11-20
> **Response to Reviewer #3**
>
> [Baselines with different number of input points]
>
> We agree that this extra ablation study would be valuable to back our claims up and strengthen the paper. We have started running separate experiments on a select subset of 40 object categories to test the performance of OccNet and PCN when trained with contexts of size 10 ~ 5000. These results should be ready by early next week. We will report them as soon as possible.
>
> [Gaussian parameterization]
>
> Although our p(h | D) is always Gaussian,  p(y | x ,D) can be made multimodal by using a more sophisticated decoder.  Indeed, generative models like VAE and GAN create multimodal distribution from Gaussian using this very mechanism.  We would also like to emphasize that, at the level of p(h|D), our model is already less restrictive than most competitors that use a deterministic decoder.
>
> [Technical novelty]
>
> We believe that our technical novelty lies in the way meta-learning can be applied to the 3D shape completion problem. In particular, as R2 agrees,  the formal treatment of sparse point cloud in the shape completion as a contextual information in the Neural Process type meta-learning method is novel.
>
> [F-score evaluation]
>
> We are currently evaluating our models using the F-Score metric. We will include these results in the final version of the paper, and will report here as soon as possible.

---

### Decision · Program_Chairs · 2021-01-07
**Final Decision**

**Decision:**

Reject

**Comment:**

This submission generated a lot of discussion.

The main strengths of the paper:
* It is an interesting application of meta-learning (to 3D shape completion), and a novel one.
* It appears to work well: it is remarkable that the proposed model can reconstruct shapes as well as it does given a point cloud with only 50 input points.

The main concern (raised by two reviewers) is that while the method is described using the language of meta-learning, the proposed architecture ends up looking very similar to a variational autoencoder: a point cloud encoder that outputs some distribution in a latent space, which is then sampled from to produce a code which drives an implicit surface decoder. The only difference appears to be that the proposed method uses a factored distribution in latent space, whereas a traditional VAE uses a non-factored one (i.e. a single multivariate Gaussian over all dimensions of the latent code).

One reviewer engaged the authors in a discussion about this point, but the resulting conversation was not satisfactory. One interpretation of the authors' response is that they are simply not aware of how a VAE could be trained using variable-sized point clouds as input (which is quite possible using many standard point cloud processing networks). However, at other points, they do seem to grasp this, when they write that "This flexible representation of the uncertainty [i.e. the one proposed by the authors] cannot be attained by the mere average or max pooling aggregation [what a PointNet encoder would do]." They even go on to provide an additional ablation study where they replace their factored probabilistic encoder with a deterministic mean/max pool encoder, and show worse results. Unfortunately, they never compare against *probabilistic* variants of such encoders (i.e. where the mean/max pool output is then used to compute a mean and variance).

Without seeing this comparison, the reviewers believe that this paper cannot be accepted, and I am inclined to agree.

On a related note: one reviewer pointed out an issue with unfair comparisons, in that baselines were trained on high density point clouds and evaluated on low density ones. The reviewer noted that these methods could be (and should have been) trained on point clouds of varying density. Perhaps this relates to my hypothesis that the authors initially did not understand that training such encoders on variable-sized point clouds was possible. In any event, in their rebuttals, they have reported some preliminary results from experiments which do this type of training, but these results are not conclusive. Complete, conclusive results from these experiments would also need to be presented before this paper could be accepted.